# Overlooked shelf sediment reductive sinks of dissolved rhenium and uranium in the modern ocean

Qingquan Hong [1], Yilin Cheng[2,3], Yang Qu[1], Lin Wei[2,3], Yumeng Liu[1], Jianfeng Gao[4], Pinghe Cai[2,3] & Tianyu Chen [1] ✉

Rhenium (Re) and uranium (U) are essential proxies in reconstructing past oceanic oxygenation evolution. However, their removal in continental shelf sediments, hotspots of early diagenesis, were previously treated as quantitatively unimportant sinks in the ocean. Here we examine the sedimentary reductive removal of Re and U and their coupling with organic carbon decomposition, utilizing the $^{224}$Ra/$^{228}$Th disequilibria within the East China Sea shelf. We identified positive correlations between their removal fluxes and the rates of sediment oxygen consumption or organic carbon decomposition. These correlations enable an evaluation of global shelf reductive sinks that are comparable to (for Re) or higher than (~4-fold for U) previously established suboxic/anoxic sinks. These findings suggest potential imbalances in the modern budgets of Re and U, or perhaps a substantial underestimation of their sources. Our study thus highlights shelf sedimentary reductive removal as critical yet overlooked sinks for Re and U in the modern ocean.

Reconstructing the oxygenation history is vital for understanding the Earth's geochemical and biological evolution[1]. Elemental enrichment and isotopic compositions of rhenium (Re) and uranium (U) in sedimentary records are valuable but still developing proxies of global ocean redox conditions for the geological past (e.g., refs. [2–9]). Rhenium and U present as soluble perrhenate (ReO$_4^-$) and uranyl carbonate ion (UO$_2$(CO$_3$)$_3^{4-}$), respectively, and behave conservatively in oxygenated seawater[10,11]. They are reduced to insoluble species and removed to sediments under suboxic/anoxic conditions (e.g., refs. [2,3,12]). Overall, their accumulation in sediments predominantly depends on the reductive removal of porewater Re and U supplied via diffusion and irrigation across the sediment–water interface[12–15], and deposition of particles (organic matter and/or disordered CaCO$_3$) that scavenge seawater Re and U[16–19]. The residence times of seawater Re (~130 kyr) and U (~500 ky) are substantially longer than the average ocean mixing timescale of ~1–2 kyr[20,21]. Hence, their element concentrations and isotopic compositions are relatively uniform across different ocean basins.

The reliable application of these Re and U proxies requires a thorough understanding of their budgets (e.g., sedimentary removal flux) and associated isotope fractionation in the modern ocean. Recent studies posited global mass balance for both elements and proposed suboxic (defined as sediments with oxygen penetration depth (OPD) < 1 cm) and anoxic (no oxygen penetration and H$_2$S ~ 0 μM) sediments as the dominant sink, although there are significant uncertainties[3,5,20]. Other sinks include coastal lagoons and blue carbon ecosystems (e.g., salt marshes, and mangrove swamps)[17,20]. Due to the scarcity of available isotope data[22–24], the Re isotope systematics in the modern ocean has been poorly evaluated. The suboxic and anoxic sinks in reaching oceanic U isotope (δ$^{238}$U) mass balance is based on operationally defined suboxic/anoxic seafloors and area-weighted mass accumulation rates of authigenic U in these areas (e.g., refs. [2,8,20]). The established sinks were based on the extrapolation of limited data (collected at sites below ~100 to thousands of meters) to the suboxic and anoxic seafloor areas at water depths >200 m [3,20]. The

[1]State Key Laboratory for Mineral Deposits Research, School of Earth Sciences and Engineering and Frontiers Science Center for Critical Earth Material Cycling, Nanjing University, Nanjing 210023, China. [2]State Key Laboratory of Marine Environmental Science, Xiamen University, Xiamen 361005, China. [3]College of Ocean and Earth Sciences, Xiamen University, Xiamen 361005, China. [4]State Key Laboratory of Ore Deposit Geochemistry, Institute of Geochemistry, Chinese Academy of Sciences, Guiyang 550081, China. ✉e-mail: tianyuchen@nju.edu.cn

shelf (water depths: ≤200 m) sediments are suboxic settings due to the high sediment and organic matter loadings. Their role in the oceanic Re and U budgets has been dismissed or treated as authigenically neutral using areal burial rates at least two orders of magnitude lower than the representative sinks in suboxic and anoxic seafloors[3,20], although shelf sediments show distinct Re and U enrichment (e.g., refs. 13,18,20,25). However, shelf sediments are disproportionately important in global budgets of trace metals, such as iron and rare earth elements[26,27]. The shelf sediment regime is a hotspot of organic carbon (OC) supply and remineralization, which limits oxygen penetration and causes anoxic conditions. This will spark the sedimentary reductive removal and authigenic accumulation of Re and U, thereby influencing the sink terms in the oceanic Re and U budgets. Recent studies, based on either the sediment accumulation or sedimentary reductive removal rates, found a shelf sediment sink to be up to 600 pmol Re m$^{-2}$ d$^{-1}$ and ~200 nmol U m$^{-2}$ d$^{-1}$ (refs. 13,14,28,29), which are higher than the representative suboxic sinks of 61.6 pmol Re m$^{-2}$ d$^{-1}$ and 1.9 nmol U m$^{-2}$ d$^{-1}$, respectively[3,20]. In view of the higher fluxes than the proposed representative suboxic/anoxic sinks, quantitative evaluation of the reductive removal of Re and U in shelf sediments provides opportunities to improve our understanding of the modern oceanic cycle of these elements and their application as oceanic redox proxies.

The East China Sea (ECS; Supplementary Fig. 1) shelf receives large suspended sediments and OC loadings, resulting in sedimentation rates of a few cm yr$^{-1}$ (ref. 30). The shelf dynamics are distinctly stronger in winter than in summer[31]: the eddy diffusivities in the bottom water are 5-fold higher; the sediments are more intensively reworked/irrigated as evidenced by deeper penetration of DO and excess $^{234}$Th activity, as well as $^{224}$Ra/$^{228}$Th disequilibria. Additionally, upwelling develops in June, reaches its strongest condition in July and August, and eventually vanishes in late September[32]. Intensive winter reworking/irrigation enhances the OC decomposition rates to ~3.5–522 mmol m$^{-2}$ d$^{-1}$ in the ECS, whereas OC decomposition is relatively low under maximal OC supply and peak seasonal hypoxia in summer[31,33]. The contrasting conditions make it an ideal region to evaluate the sedimentary reductive removal of Re and U, their relation with OC decomposition, and the role of shelf sediments in the global oceanic Re and U budgets.

In shelf sediments, irrigation, which is driven by the flushing of burrows and coupled with the diffusion of solutes between burrows and the surrounding sediments[34,35], acts to supply solutes to sediment porewater when solute concentrations in bottom water exceed porewater levels. Previous studies highlight the importance of irrigation in augmenting the sedimentary reductive removal of Re and U in shallow water sediments[14,36]. In the ECS, irrigation has been shown to enhance the benthic DO, NH$_4^+$, Fe, and Be fluxes[31,37,38]. The $^{224}$Ra/$^{228}$Th disequilibrium method has been proven suitable for flux estimation over a timescale of ~10 days in highly dynamic systems influenced by sediment reworking/irrigation[31,39–41]. It is based on the different geochemical behavior between the dissolvable $^{224}$Ra (half-life: 3.66 days) and its highly particle-reactive parent $^{228}$Th (half-life: 1.91 years) in the marine environment[37,42]. Processes like diffusion, bioturbation, irrigation, and reworking render the transfer of dissolved $^{224}$Ra between sediment and the overlying water, thereby resulting in a deficit of $^{224}$Ra relative to $^{228}$Th that can be integrated to derive the benthic flux of $^{224}$Ra. This flux can be used to quantify the sedimentary reductive removal rates of Re and U based on the ratio of the concentration gradients between dissolved $^{224}$Ra and Re (U). A key advantage of this $^{224}$Ra/$^{228}$Th disequilibrium method over traditional sediment incubation and modeling methods is that it captures all physical processes that affect solute transfer across the sediment–water interface but does not alter the system[31,39–41].

In this work, we measure the porewater Re and U concentrations in the shelf sediments from the ECS during the summer and winter seasons. Subsequently, we apply the $^{224}$Ra/$^{228}$Th disequilibrium method to estimate the sedimentary reductive removal fluxes of Re and U. Our analysis reveals significant positive correlations between these fluxes and the sediment oxygen consumption rate. Our calculation suggests sedimentary reductive removal processes in continental shelf environments as critical sinks affecting the modern oceanic budgets of Re and U. These findings imply the sources and sinks of Re and U in the modern ocean may be imbalanced and/or their sources are substantially underestimated.

## Results and discussion
### Porewater Re and U concentrations
Porewater geochemistry of DO, nutrients, and trace metals are shown in order to depict the redox conditions regulating sedimentary reductive removal of Re and U. The dataset of $^{224}$Ra/$^{228}$Th, DO, nutrients, and Fe for the cruises of 2017 and 2018 are available from previous studies[31,43]. New data includes $^{224}$Ra/$^{228}$Th, nutrients, and Fe for the samples collected in the summer of 2021 and RSMs (Mn, Re, and U) for all cruises. These datasets are shown in Fig. 1 (Re, U, and $^{224}$Ra/$^{228}$Th ratio) and Supplementary Fig. 2 (Fe, Mn, and nutrients) and compiled in Supplementary Tables 1 and 2. Here, we focus on the sedimentary reductive removal fluxes and their role in the modern oceanic budgets of Re and U.

Porewater Re and U concentrations ([Re]$_{diss}$ and [U]$_{diss}$) fall in the range of 2.7–115 pM and 0.3–27.3 nM, respectively. [Re]$_{diss}$ and [U]$_{diss}$ in the upper ~1–2 cm of some sites (11 and 8 out of 17 cores, for Re and U, respectively) are higher than the seawater values (Re: 40.3 ± 2.4 pmol L$^{-1}$, U: 10.7 ± 1.0 nmol L$^{-1}$; $N$ = 23, 1 SD; Supplementary Fig. 3). Beneath these layers, [Re]$_{diss}$ and [U]$_{diss}$ are lower than those in the seawater. They decrease smoothly and reach asymptotic minima comparable to those in other margins (~3–14 pM and ~2–4 nM, respectively)[14,18,44]. Incomplete removal of Re and U from porewater may be associated with a colloidal phase[12] and/or a minimum U concentration required for microbial-mediated removal[45]. However, not all stations show the typical removal pattern. Broad maxima of [Re]$_{diss}$ and [U]$_{diss}$ extend to ~6–10 cm at St. B3-W, E4-S, F1-S, and Y7-S.

Sedimentary removal of Re and U is strongly associated with reducing conditions in marine sediments[4]. In the ECS, DO exhausts within the upper 0.12–0.69 cm across all sites, and reducing conditions favoring metal oxides and sulfate reduction dominate the OC decomposition[31]. During summer, [Re]$_{diss}$ and [U]$_{diss}$ generally decrease slowly, together with slow increases in terminal metabolism product of anaerobic OC decomposition ([NH$_4^+$]$_{diss}$, [Mn]$_{diss}$, [Fe]$_{diss}$) and less decline of SO$_4^{2-}$/Cl ratios. On the contrary, [Re]$_{diss}$ and [U]$_{diss}$ progressively decrease to asymptotic minima at relatively shallower depths in winter (Fig. 1a, b and Supplementary Fig. 2). The stronger OC decomposition and more reducing conditions, evidenced by lower SO$_4^{2-}$/Cl ratios, enhanced [NH$_4^+$]$_{diss}$, and [Fe]$_{diss}$ maxima, are probably responsible for the relatively faster Re and U removal in winter. The change of gradients and the minima of [U]$_{diss}$ coincide with the broad maxima of [Fe]$_{diss}$. Re reduction and removal start from the depth of NO$_3^-$ and MnO$_2$ reduction and could extend to the bottom layer conducive to Fe oxides and sulfate reduction.

Larger $^{224}$Ra deficits ($^{224}$Ra/$^{228}$Th ratio <1) in deeper depth intervals were observed at most sites collected in winter (Fig. 1c), revealing more intense irrigation/reworking of porewater, which was also evidenced by deeper penetration of oxygen and excess $^{234}$Th into the sediments[31].

### Sources of Re and U for sedimentary reductive removal
The elevated porewater Re and U concentrations at the sediment–water interface of some sites exceed those found in the seawater (Fig. 1a, b), indicating additional sources supplying porewater Re and U for reductive removal, apart from irrigation and diffusion of dissolved species from bottom seawater (Supplementary Fig. 5)[12–15,18]. Under oxic bottom water, enhanced bioturbation may result in the re-oxidation of reduced Re and U[14,18]. However, the most distinct

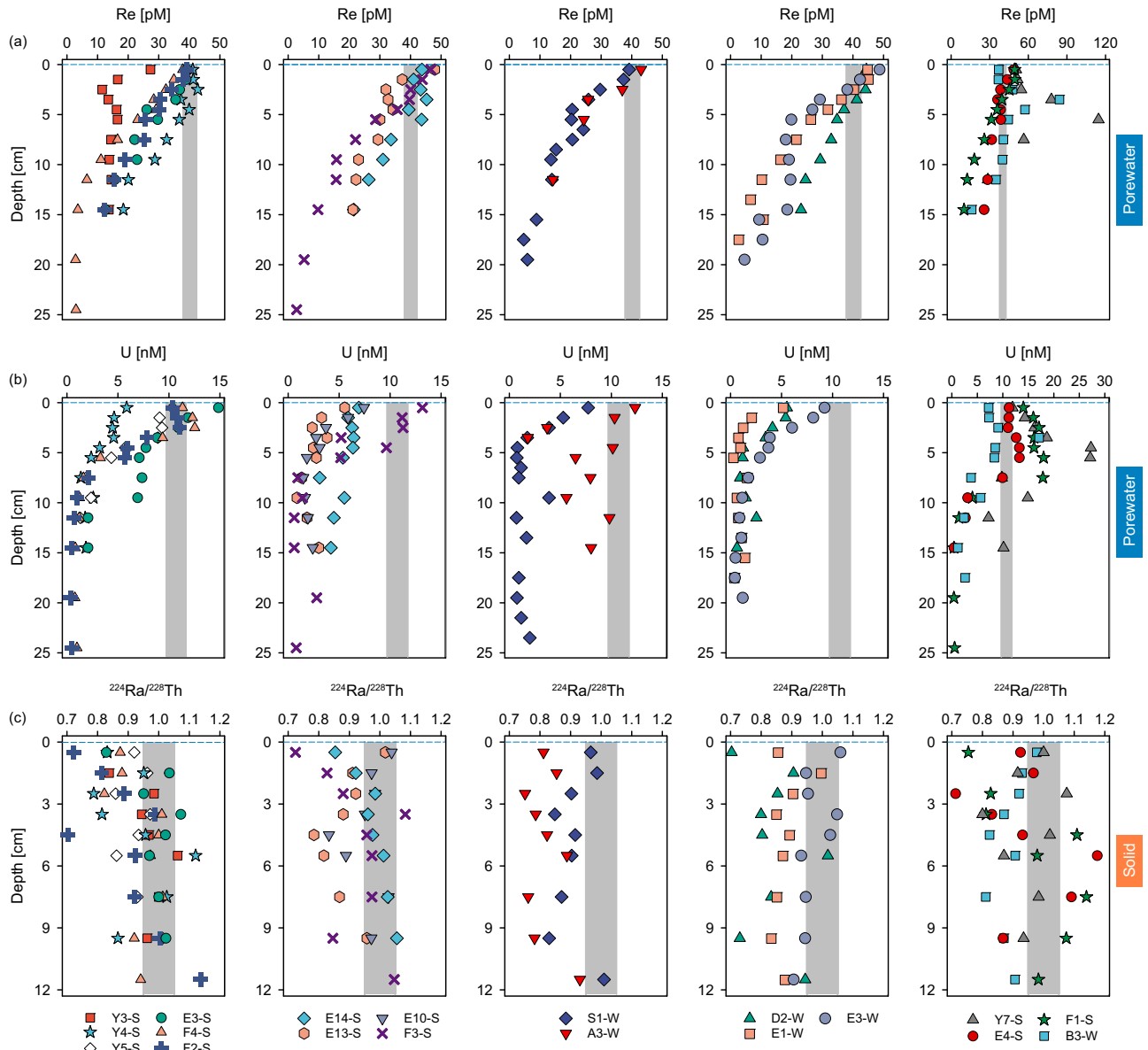

**Fig. 1 | Porewater Re and U, as well as the bulk $^{224}$Ra/$^{228}$Th disequilibria in the shelf sediments of the East China Sea.** Vertical profiles of Re and U in porewater (**a**, **b**) and the bulk $^{224}$Ra/$^{228}$Th disequilibria in the sediments (**c**). The samples were collected within 10 days during the summer of 2017 and 5 days during the summer of 2021 and the winter of 2018. The horizontal dash line represents the sediment–water interface. The vertical bar in the panels of Re and U represents the average concentrations±1 SD ($N$ = 23) measured in the water column of the East China Sea (see Supplementary Fig. 3). The vertical bar in the $^{224}$Ra/$^{228}$Th panels denotes the overall uncertainty of the $^{224}$Ra/$^{228}$Th ratios, with values lower than the left boundary denoting the release of $^{224}$Ra from sediments.

enrichments of core-top porewater [Re]$_{diss}$ and [U]$_{diss}$ are observed in summer with lower bioturbation compared to winter (Fig. 1). In this regard, re-oxidation of reduced Re and U is unlikely to be the dominant source for the observed pattern at these sites. Instead, the decomposition and transformation from freshly deposited particles (particularly organic matter) at the sediment–water interface might have released relatively large amounts of Re and U (i.e., regenerated flux) compared to the reductive removal[14,16,18,19]. If the "particulate source" scavenged from is not quantitatively removed by sedimentary reductive removal, it could even result in a dissolved flux back to the water column (Supplementary Fig. 5).

Moreover, the broad maxima of [Re]$_{diss}$ and [U]$_{diss}$ at St. B3-W, E4-S, F1-S, and Y7-S could not be supported merely by their release at the sediment–water interface[31]. Alternatively, they may be indicative of enhanced Re and U release from the solid phases into porewater at these depth intervals, probably due to intense sediment reworking.

Below these depths, Re and U are steadily removed to reach the typical asymptotic minima.

Regardless of the routes (dissolved versus particulate) that supply Re and U to the porewaters, it is the reductive removal that ultimately results in the enrichment of Re and U in the modern shelf sediments. Our data of Re and U concentrations in the corresponding solid phases provides further support for the reductive enrichment process (see Supplementary Text 1 and Supplementary Fig. 4). The $^{224}$Ra/$^{228}$Th disequilibria approach, which incorporates the influence of irrigation in regulating solute fluxes, is used to estimate the reductive removal fluxes of Re and U ($F_i$, positive values denote removal)[37]. A detailed description of the calculation is provided in Methods.

## Sedimentary reductive removal fluxes of Re and U

The total Re and U fluxes vary from 20 ± 5 to 4600 ± 950 (median: 320 ± 61) pmol m$^{-2}$ d$^{-1}$ and from 1.5 ± 0.6 to 830 ± 320 (median: 36 ± 13)

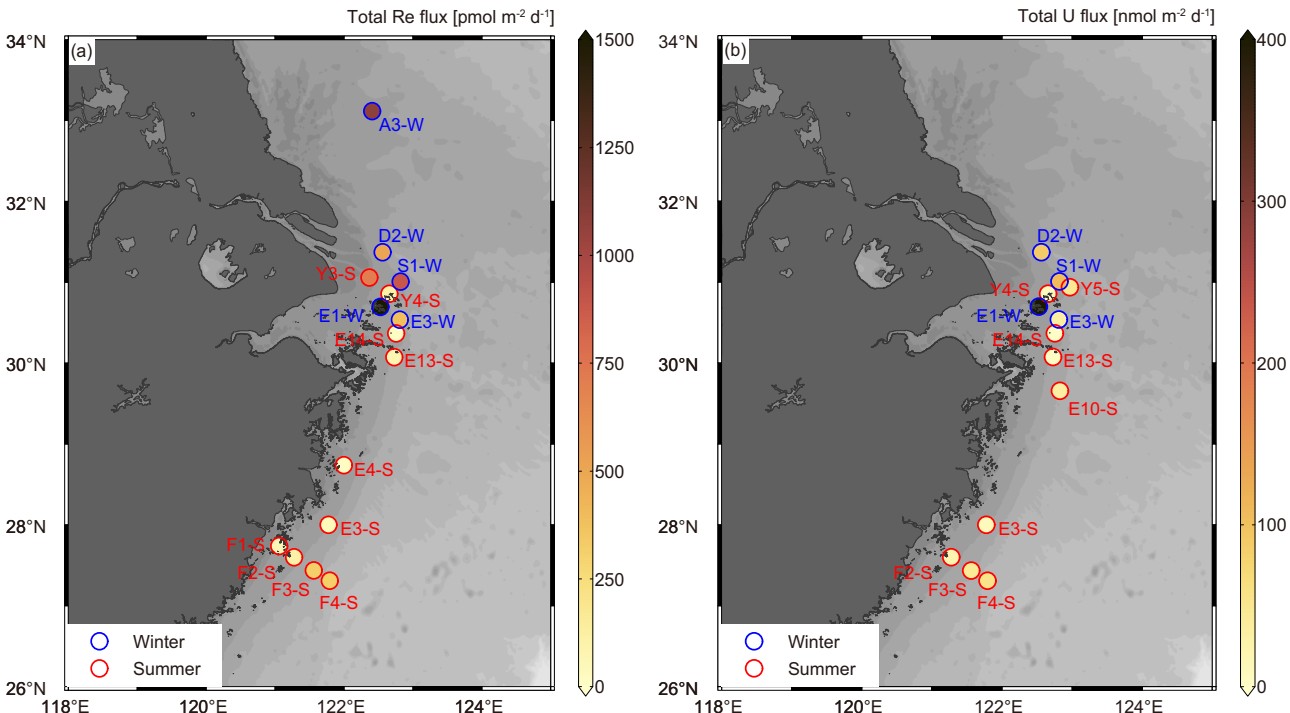

**Fig. 2 | Spatial variation of the sedimentary reductive removal fluxes of Re and U in the East China Sea. a**, **b** Fluxes of Re and U, respectively. The stations surveyed in winter and summer are marked with blue and red circles, respectively. Figures were produced using Ocean Data View[70].

nmol m$^{-2}$ d$^{-1}$, respectively (Fig. 2 and Table 1). They are ~15 times higher than the diffusive fluxes (7.0−66 pmol Re m$^{-2}$ d$^{-1}$ and 0.6−3.7 nmol U m$^{-2}$ d$^{-1}$, respectively; Table 1) at the same stations, suggesting remarkable enhancement of sedimentary Re and U removal by irrigation. The amplification factor of sediment area (see the Methods), a dimensionless parameter, is used to characterize the enhancement of solute exchange driven by irrigation relative to that driven by molecular diffusion[31]. Unexpectedly high removal fluxes of Re and U were observed at St. E1-W located near a refuge harbor and probably driven by accidental anthropogenic disturbance, as indicated by the extreme amplification factor of sediment area. These fluxes would be similar to the other cores (1200 ± 250 pmol Re m$^{-2}$ d$^{-1}$ and 220 ± 84 nmol U m$^{-2}$ d$^{-1}$, respectively) and fall much closer to the trend if the amplification factor decreases by ~4 times to the average of the other cores in winter. In the following discussion, this station is omitted for a more representative estimation of global shelf sedimentary reductive sinks.

The amplification factor of sediment area in the ECS is higher than that estimated with the sediment chamber incubation method in the North American marginal sediments (NAMs: ~3)[14,36]. This might be partly caused by the temporal and spatial variability between these studies. Another plausible explanation is that the alteration of the physical conditions during sediment incubation may dismiss small-scale advective processes and thus underestimate the fluxes[39,43]. The sedimentary reductive removal rates of Re and U in the ECS are comparable to the accumulation rates in shelf sediments with well-oxygenated water columns in the NAMs and the East Siberia Sea (8−600 pmol Re m$^{-2}$ d$^{-1}$ and 4.6−106 nmol U m$^{-2}$ d$^{-1}$; refs. 13,28,36,46).

The sedimentary reductive removal rates show remarkable seasonal variation (Fig. 2 and Table 1), with lower fluxes measured in summer (average: ~180 pmol Re m$^{-2}$ d$^{-1}$ and 26 nmol U m$^{-2}$ d$^{-1}$) when OC supply and seasonal hypoxia are at maximum, and the seafloor is relatively stagnant. In contrast, they are ~3−4 times higher (average: ~720 pmol Re m$^{-2}$ d$^{-1}$ and 78 nmol U m$^{-2}$ d$^{-1}$) in winter under well-oxygenated water column and rough seafloor conditions. It is at odds with the expectation that these fluxes would culminate during summer

when the maximum OC supply and the upwelling promote the development of hypoxia[47] and, consequently, the sedimentary reductive removal of RSMs[2,3]. Sedimentary OC decomposition dictates the reducing conditions conducive to the removal of Re and U[13,14]. Intense winter reworking/irrigation was posited to be responsible for the highly efficient decomposition of sedimentary OC in the study area, resulting in relatively homogeneous TOC contents (0.33 ± 0.05%) and stronger reducing conditions[31]. Therefore, we argue that the distinctly higher winter removal rates of Re and U are driven by intensive reworking/irrigation. It is evidenced by the positive correlations between the removal fluxes of Re and U and the sediment area amplification factors (R$^2$ = 0.59 and 0.73 for Re and U, respectively, Fig. 3a, b). Irrigation brings DO, labile OC, Re, and U to sediments. The supply of labile OC overwhelms DO in controlling the sediment redox conditions. Therefore, intensive winter reworking/irrigation would trigger the development of more vigorous anaerobic diagenesis and thus enhance the reductive removal of Re and U (see next section). Overall, the observation suggests that the sedimentary reductive removal fluxes of Re and U in shelf sediments are not regulated by water column oxygenation conditions and are significantly higher than the prior estimates (e.g., 0.24 pmol Re m$^{-2}$ d$^{-1}$; refs. 3,20).

### Regenerated flux of Re and U out of the sediment−water interface

The regenerated upward fluxes of Re and U estimated with our pore-water data range from −860 to 0 pmol m$^{-}$ d$^{-1}$ (median: −49 pmol m$^{-2}$ d$^{-1}$; average: −223 pmol m$^{-2}$ d$^{-1}$) and −40 to 0 nmol m$^{-2}$ d$^{-1}$ (median: 0 pmol m$^{-2}$ d$^{-1}$; average: −6 pmol m$^{-2}$ d$^{-1}$), respectively (Table 1). For some sites, the regenerated Re and U fluxes are comparable to or even larger than the sedimentary reductive removal fluxes. Regarding the median values, they are ~15% and 0% of the sedimentary reductive removal fluxes of Re and U, respectively, since many sites have zero upward fluxes. Regarding the average values, the regenerated upward fluxes could be ~36% and ~6% of the reductive removal fluxes of Re and U at our studied sites, respectively. It suggests that the regenerated flux

**Table 1 | Fluxes of $^{224}Ra$, DO, $NH_4^+$, Re and U, and amplification factors of sediment surface area in the East China Sea**

| Total fluxes based on the $^{224}Ra/^{228}Th$ disequilibria approach | | | | | | | | Diffusive flux | |
|---|---|---|---|---|---|---|---|---|---|
| Station | DO $\mu mol\ L^{-1}$ | $\xi$ [b] | $^{224}Ra$ [b] $dpm\ m^{-2}\ d^{-1}$ | DO [b] $mmol\ m^{-2}\ d^{-1}$ | $NH_4^+$ [b] $mmol\ m^{-2}\ d^{-1}$ | Re [a] $pmol\ m^{-2}\ d^{-1}$ | U [a] $nmol\ m^{-2}\ d^{-1}$ | Re $pmol\ m^{-2}\ d^{-1}$ | U $nmol\ m^{-2}\ d^{-1}$ |
| *Summer 2017* [b] | | | | | | | | | |
| Y3-S | 65 | 11 | 730 ± 110 | 30 ± 4 | 10 ± 1 | 720 ± 210 (0) | –[c] | 66 ± 13 | –[c] |
| Y4-S | 116 | 11 | 990 ± 170 | 59 ± 20 | 13 ± 2 | 110 ± 23 (−49 ± 9) | 11 ± 2.4 (0) | 11 ± 1.0 | 1.1 ± 0.1 |
| Y5-S | 76 | 26 | 1000 ± 160 | 35 ± 10 | 8.8 ± 1.6 | –[c] | 49 ± 12 (0) | –[c] | 1.9 ± 0.3 |
| Y7-S | 115 | 25 | 740 ± 190 | 45 ± 14 | 4.6 ± 1.2 | –[d] | –[d] | –[d] | –[d] |
| E14-S | 84 | 1.9 | 270 ± 55 | 6 ± 1 | 1.6 ± 0.4 | 20 ± 5.1 (−41 ± 9) | 1.5 ± 0.6 (0) | 11 ± 1.4 | 0.8 ± 0.2 |
| E13-S | 118 | 10 | 890 ± 110 | 10 ± 3 | 3.6 ± 0.5 | 82 ± 19 (−400 ± 56) | 6.3 ± 1.8 (0) | 8.5 ± 1.5 | 0.6 ± 0.2 |
| E10-S | 144 | 15 | 450 ± 170 | 29 ± 11 | 3.7 ± 1.4 | –[c] | 34 ± 14 (0) | –[c] | 2.6 ± 0.2 |
| E4-S | 136 | 3.8 | 300 ± 180 | 17 ± 10 | 2.8 ± 1.7 | 26 ± 16 (−150 ± 93) | –[d] | 7.3 ± 1.1 | –[d] |
| E3-S | 164 | 3.7 | 220 ± 85 | 25 ± 10 | 4.5 ± 1.7 | 53 ± 22 (0) | 13 ± 5.5 (−30 ± 13) | 15 ± 1.4 | 3.7 ± 0.4 |
| *Summer 2021* | | | | | | | | | |
| F1-S | 169 | 3.8 | 490 ± 170 | 9 ± 5 | 5.6 ± 1.9 | 65 ± 23 (−180 ± 62) | –[d] | 25 ± 1.0 | –[d] |
| F2-S | 120 | 19 | 140 ± 220 | 54 ± 6 | 8.6 ± 1.5 | 120 ± 22 (0) | 22 ± 4.6 (0) | 11 ± 0.6 | 2.0 ± 0.2 |
| F3-S | 188 | 25 | 850 ± 150 | 120 ± 29 | 16 ± 3 | 320 ± 66 (−660 ± 130) | 44 ± 13 (−40 ± 8) | 20 ± 1.2 | 2.8 ± 0.6 |
| F4-S | 173 | 35 | 940 ± 140 | 50 ± 10 | 13 ± 2 | 320 ± 61 (0) | 56 ± 14 (0) | 15 ± 0.6 | 2.6 ± 0.4 |
| *Winter 2018* [b] | | | | | | | | | |
| A3-W | 241 | 129 | 4400 ± 200 | 250 ± 30 | 36 ± 4 | 1100 ± 260 (−620 ± 54) | –[d] | 9 ± 2.1 | –[d] |
| B3-W | 242 | 175 | 3800 ± 350 | 520 ± 70 | 58 ± 8 | –[d] | –[d] | –[d] | –[d] |
| D2-W | 214 | 76 | 4800 ± 360 | 220 ± 20 | 21 ± 2 | 510 ± 64 (−860 ± 90) | 91 ± 19 (0) | 7.0 ± 0.5 | 1.2 ± 0.2 |
| S1-W | 208 | 40 | 2200 ± 290 | 210 ± 40 | 26 ± 4 | 900 ± 180 (0) | 110 ± 20 (0) | 24 ± 2.6 | 2.8 ± 0.2 |
| E3-W | 190 | 13 | 840 ± 290 | 44 ± 16 | 22 ± 8 | 380 ± 140 (−390 ± 140) | 36 ± 13 (−4 ± 1) | 32 ± 1.8 | 3.0 ± 0.3 |
| E1-W | 253 | 333 | 4700 ± 450 | 680 ± 140 | 57 ± 11 | 4600 ± 950 (0) | 830 ± 320 (0) | 14 ± 0.7 | 2.6 ± 0.9 |

[a]The regenerated fluxes out of the sediments ($F_{out}$ with negative values) due to the additional release of Re and U from the "particulate source" are presented in parentheses.
[b]Fluxes of $^{224}Ra$, DO, and $NH_4^+$, as well as $\xi$ for the cruises in 2017 and 2018, are adopted from refs. 31,43.
[c]No data due to limited sample size;
[d]Fluxes cannot be calculated due to the complicated removal-release within the upper few cm indicated by the distinct zig-zag patterns.

could be an important yet not well understood source for the reductive removal of Re and U within the sediments. However, unlike rare earth elements whose riverine particle-bound phases might be actively involved in the shelf cycling (e.g., refs. 26,48,49), the regenerated upward fluxes of Re and U are most likely scavenged from seawater, as discussed above. Therefore, they are considered part of oceanic internal recycling. In this regard, we have not included these fluxes as external sources when discussing the oceanic Re and U mass balance. Even if some of the upward fluxes are from riverine particle-bound phases and/or re-oxidation of previously reduced species, our conclusion on the quantitative importance of shelf sediment sinks to the modern oceanic budgets of Re and U remains unchanged. Future studies should aim to better characterize the nature of Re and U released at the sediment–water interface, for example, by combined analyses of their isotope signatures and exchange flux.

**Sedimentary reductive removal of Re and U coupled with organic carbon decomposition**

In contrast to the $NH_4^+$ fluxes that increase exponentially with DO in the bottom water[31], no significant correlations of total Re and U fluxes with DO in bottom water and its penetration depth in sediments (Supplementary Fig. 6) were observed, suggesting a minor role of DO levels in regulating the reductive removal of Re and U. This is in line with a small impact of bottom-water anoxia on OC decomposition and burial[50]. Sedimentary OC decomposition, the primary mechanism dictating the sedimentary reducing conditions, must be governing the Re and U fluxes in the $O_2$-depleted sediments. Accordingly, stronger OC decomposition would accelerate the production and accumulation of $NH_4^+$ in porewater and enhance the DO consumption and reductive removal of Re and U from porewater[4,14]. In support of the assertion are the positive correlations between the reductive removal fluxes of Re and U and the OC decomposition rates (determined as the net $NH_4^+$ production rates; $R^2 = 0.75$ and 0.62 for Re and U, respectively, $p < 0.005$; Fig. 3c, d), as well as the total sediment oxygen consumption rates ($R^2 = 0.61$ and 0.72 for Re and U, respectively, $p < 0.0001$; Fig. 3e, f).

It is counterintuitive that sedimentary reductive removal of Re and U correlates simultaneously with sediment oxygen consumption and anaerobic OC decomposition rates. However, it is unsurprising since oxygen consumption and Re/U reduction are synergistic rather than competing during OC decomposition within shallow sediments. Microbial metal respiration coupling with sediment reworking/irrigation and anaerobic conditions has been posited to be responsible for the loss of OC in shallow sediments or at the sediment–water interface in the ECS[51,52]. In these sediments, the oxygen supply is insufficient to fully decompose the OC. Therefore, the higher sediment oxygen consumption fluxes mean a greater demand for electron acceptors[53]. Intense sediment reworking/irrigation introduces large amounts of labile OC into sediments, exposing anoxic sediments and buried recalcitrant OC. This process produces and accumulates bacterial biomass, whose remineralization would prime high anaerobic OC decomposition rates and the reduction of metal oxides and sulfate[34,36,52,54–56]. Under this circumstance, OC decomposition triggers anoxic conditions conducive to the reduction of Re and U and shoals the depth of their reductive removal[4,44,57,58]. Such phenomena are evidenced by lower $SO_4^{2-}/Cl$ ratios and elevated $[NH_4^+]_{diss}$ and $[Fe]_{diss}$ maxima in winter (Fig. 1 and Supplementary Fig. 2). Moreover, intense winter reworking/irrigation would compensate for the Re- and U-deficit and increase their availability in porewater, fueling their sedimentary reductive removal rates. Therefore, we suggest that

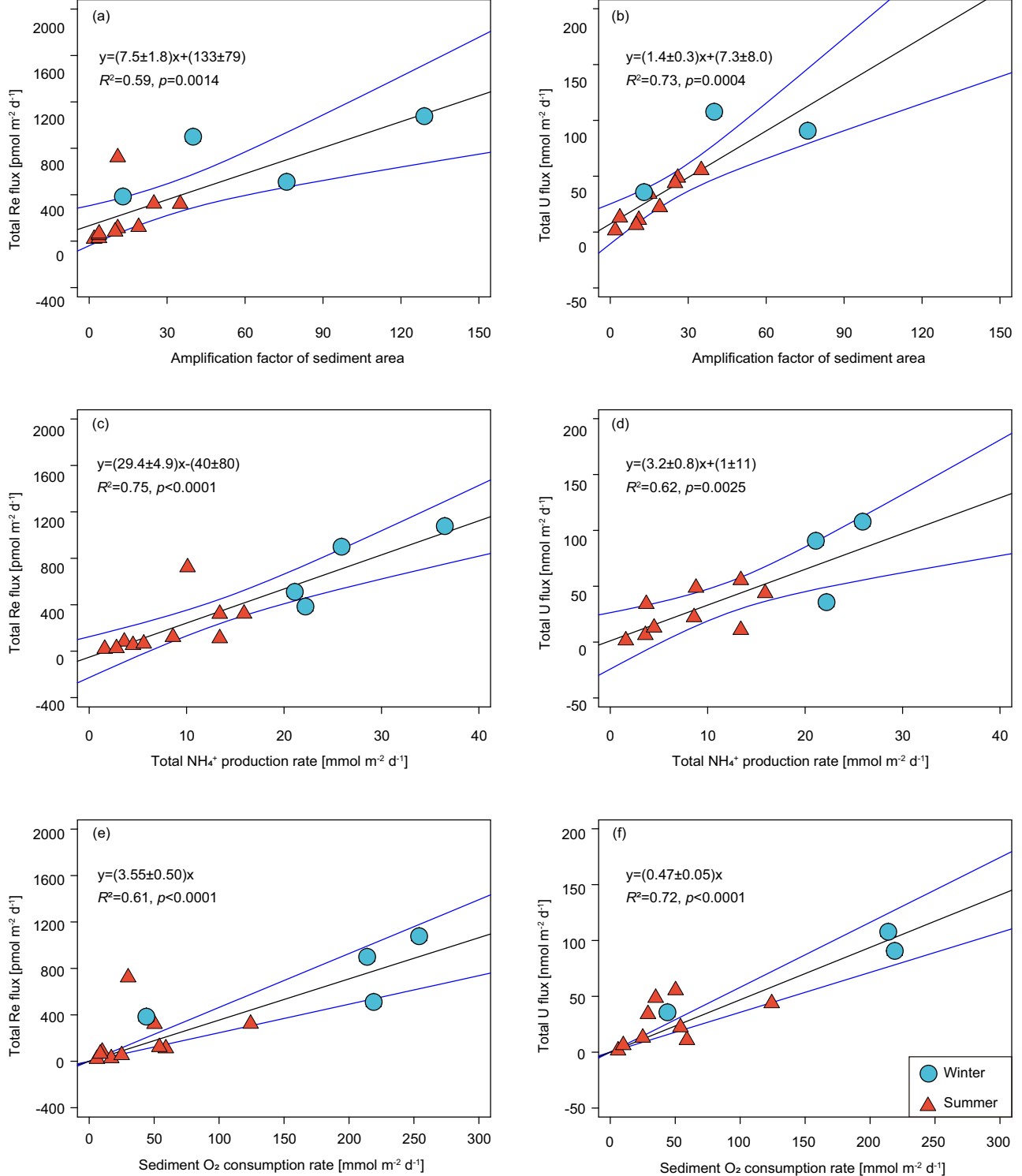

**Fig. 3 | Correlations of sedimentary reductive removal of Re and U with sediment geochemical parameters.** Correlations between sedimentary reductive removal of Re and U and amplification factor of sediment area (**a**, **b**), the anaerobic decomposition rate of organic carbon (determined as the net ammonium production; **c**, **d**), and sediment oxygen consumption rate (**e**, **f**) in the East China Sea. Triangles and circles represent data collected in summer and winter, respectively. The anomalous fluxes at St. E1-W have been excluded from the linear regression.

intensive OC decomposition, which triggers severe reducing conditions, is responsible for the strong Re and U removal in the shelf sediments. Note that the correlations between the reductive removal fluxes of Re and U and the oxygen consumption rates should only apply in shelf sediments, where DO is exhausted within 1 cm and reducing conditions prevail. In the deep ocean, OC remineralization in

the water column and sufficient DO supply suppresses the development of reducing conditions within sediments, thus hindering the reductive removal of Re and U.

A similar correlation between sedimentary reductive removal of Re and OC decomposition was observed in the NAMs sediments, overlain by oxygenated bottom waters[14,36], suggesting a potentially

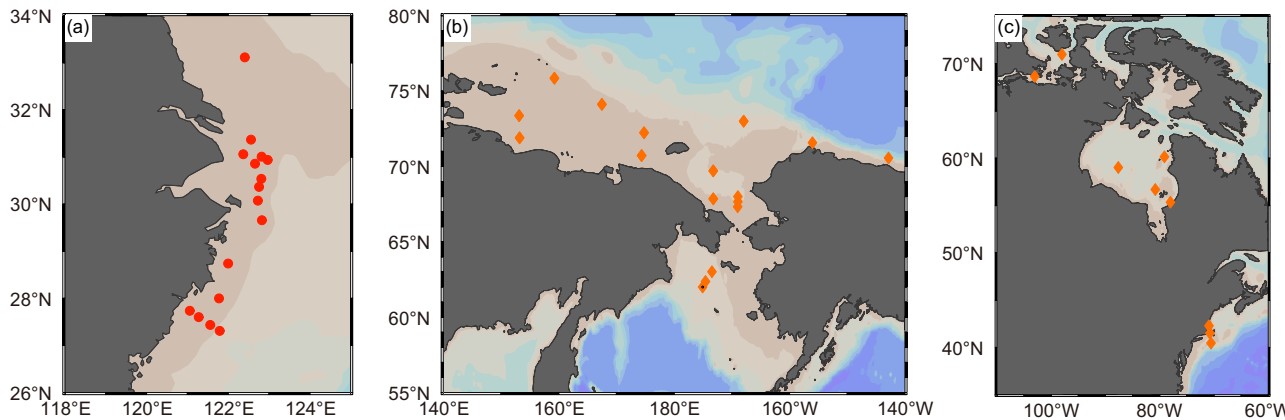

**Fig. 4 | Locations of the sedimentary removal fluxes of Re and U compiled from the global shelf. a** East China Sea (this study). **b** East Siberian Sea and Western North American margin[13,46]. **c** Eastern North American margin[13–15,28,36]. Figures were produced using Ocean Data View[70].

universal linkage between these two processes in shelf sediments. The positive correlations in both ECS and NAMs are consistent with simultaneous increases of Re accumulation and anaerobic OC decomposition[14]. Notably, the slopes of Re removal vs. anaerobic OC decomposition rates varied between these two studies ($29.4 \pm 4.9$ in the ECS vs. $10.6 \pm 2.6$ in the NAMs). A plausible explanation is the influence of stronger irrigation in the ECS, which could introduce labile OC, trigger more vigorous anaerobic diagenesis, and shoal the depth of Re removal, thus enhancing sedimentary reductive removal of Re. In the NAMs, Re removal was coincident with Fe(III) and sulfate reduction[14], while Re removal in the ECS began from the depth of $NO_3^-$ and $MnO_2$ reduction and could extend to the bottom layer conducive to Fe oxides and sulfate reduction (Fig. 1 and Supplementary Fig. 2). As a result, a higher slope is expected on the ECS shelf. Furthermore, the correlations in these two studies are based on limited data that might bias the in-between slopes. We, therefore, conclude that the OC decomposition rate should be a good reference for the reductive removal of Re and U in shelf sediments, although more data on their relationship across various environmental and geomorphic conditions is warranted.

### Shelf sediments as a critical sink of global oceanic Re and U

For the evaluation of the sedimentary reductive removal fluxes of Re and U over global shelves, the ECS, the NAMs, and the East Siberian Sea (Fig. 4) can be adopted as representative settings since they are among the widest shelves worldwide and characterized by diagenetic conditions (sedimentation rate, and OC supply/decomposition) close to the representative conditions of global shelves (Supplementary Table 3). The compiled flux of Re from these regions (median: 112 pmol m$^{-2}$ d$^{-1}$; 95% confidence interval (CI): [120, 299] pmol m$^{-2}$ d$^{-1}$; Supplementary Table 4) is comparable to the sinks in suboxic and anoxic seafloor (62 and 197 pmol m$^{-2}$ d$^{-1}$, respectively; Fig. 5a). A simple extrapolation of the compiled flux to the global suboxic sediment (area: $1.68 \times 10^7$ km$^2$; ref. 3) yields a suboxic sink approximately 1.6 times the dominant Re source from rivers[21] (median [95% CI]: 685 [734, 1830] Kmol yr$^{-1}$; Table 2). For U, the compiled flux (median: 30 nmol m$^{-2}$ d$^{-1}$; 95% CI: [25, 41] nmol m$^{-2}$ d$^{-1}$; Supplementary Table 4) is much higher than the representative suboxic flux in sediments below 200-m water depth and comparable to the anoxic sink (Fig. 5b). If extrapolated to the global suboxic sediments, it yields a suboxic sink of 184 (95% CI: [154, 252]) Mmol yr$^{-1}$, which is an order of magnitude higher than previous estimates (Fig. 6 and Table 2; refs. 5,20).

It is worth noting that the reductive removal rates of Re and U in suboxic seafloor sediments are expected to drop with water depth due to the less reducing conditions associated with lower OC contents and DO consumption[13,14,53]. Because of the heterogeneity of the fluxes on

the global shelves (Supplementary Table 4), simply extrapolating the compiled fluxes to global suboxic sediments may bias the sinks toward the high values of shallow water regions and result in significant uncertainties. We have identified strong correlations between the reductive removal of Re and U and the sediment oxygen consumption due to their synergistic effect during OC decomposition (see last section). The sediment oxygen consumption has been well constrained over the global ocean, with mean areal rates of $25 \pm 13$ and $9 \pm 5$ mmol m$^{-2}$ d$^{-1}$ on the global inner (10–50 m) and outer (50–200 m) shelves, respectively[53]. Thus, a more reasonable evaluation of the global shelf sinks of Re and U with narrower uncertainties can be made by taking advantage of the identified correlations between Re and U fluxes and the sediment oxygen consumption rates. Here, the flux ratios between Re and U uptake and sediment oxygen consumption with zero intercept ($3.55 \pm 0.50$ pmol mmol$^{-1}$ for Re and $0.47 \pm 0.05$ nmol mmol$^{-1}$ for U, respectively; Fig. 3e, f) are used because there would be no Re and U removal when no oxygen consumption occurs, i.e., no oxidant is required. By upscaling the flux ratios to the global shelves, a Re sink of $431 \pm 175$ Kmol yr$^{-1}$ can be calculated over global shelf sediments (water depths: 10–200 m). It is comparable to the suboxic sink beyond the shelf and the dominant source from rivers (Fig. 6 and Table 2; ref. 3). Accordingly, the Re sink over the global suboxic seafloor sediments can be conservatively updated to $806 \pm 175$ Kmol yr$^{-1}$ (Fig. 6 and Table 2), roughly consistent with the lower bound of the shelf sink estimated using the extrapolation of the compiled median areal flux. A similar calculation yields a U sink of $57 \pm 22$ Mmol yr$^{-1}$ over global shelves, which is approximately three times the sum of anoxic and suboxic sinks beyond the shelf and still exceeds the total U input[5,20].

Based on these updated constraints on the shelf sediment sinks using the correlations between the reductive removal of Re and U and the sediment oxygen consumption, both the Re and U budgets are imbalanced, i.e., the sink is ~80–100% larger than the source (Table 2). Therefore, the residence time of Re and U, expressed as the ratio of the reservoir to the output flux ($\tau = M/F_{out}$; ref. 59), would be much lower than previous estimates. Possible additional sources or glacial-interglacial cycles of sinks should be invoked to account for the imbalanced budgets of oceanic Re and U. From the source perspective, the lack of a temporal-spatial integration of the rivers may noticeably bias the dominant source, although prior estimates compiled Re and U concentrations from rivers encompassing 37% and 60% of total water runoff to derive this input[20,21]. Particle disaggregation/dissolution during estuarine mixing could enhance the overall fluxes of trace metals from rivers[60]. For example, it could enhance the Re and U fluxes from the Amazon River system by ~4–5 times[12,61], and enhance the Re fluxes by ~30% and ~15% in the Mississippi River delta and the Jiulong

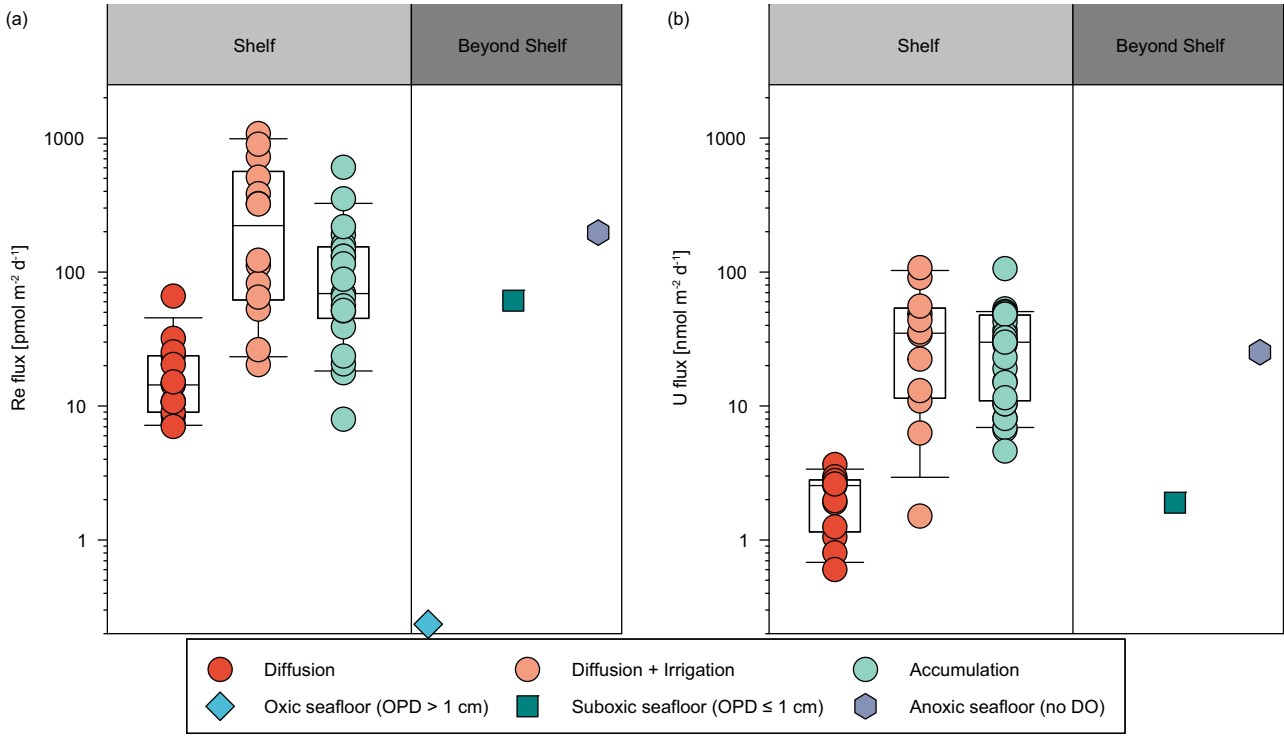

**Fig. 5 | Compilation of areal sedimentary reductive removal fluxes of Re and U.** Areal sedimentary removal fluxes of Re (**a**) and U (**b**) are split into two groups: shelf (gray) vs beyond shelf (dark gray), respectively. The compiled fluxes include the diffusive fluxes estimated from porewater profiles, the total fluxes derived from the $^{224}$Ra/$^{228}$Th disequilibria approach (this study), and authigenic solid phases in shelf sediments from refs. 13–15,28,36,46, and the representative accumulation rates used for oxic (oxygen penetration depth (OPD) > 1 cm), suboxic (OPD ≤ 1 cm), and anoxic (no dissolved oxygen (DO)) settings beyond shelf[3,20]. Error bars above and below the boxes indicate the 90th and 10th percentiles. The black horizontal lines of the boxes represent the 75th, 50th, and 25th percentiles.

**Table 2 | First-order estimate of modern oceanic Re and U budgets based on the updated shelf sediment sink**

|  | Re [Kmol y$^{-1}$] | % of input flux | U [Mmol y$^{-1}$] | % of input flux | δ$^{238}$U [‰] |
|---|---|---|---|---|---|
| Total input | 429 |  | 53 |  |  |
| River dissolved load | 429[a] | 100% | 42[b] | 79% | −0.34[c] |
| Groundwater | – | – | 9.3[b] | 18% | – |
| Aeolian | – | – | 1.8[b] | 3.4% | – |
| Total output (Updated Suboxic+Anoxic+Oxic) | 860 ± 175 | 200 ± 41 | 99.4 ± 27.2 | 188 ± 51 | −0.31 ~ −0.30 |
| Anoxic sediment | 28.0[d] | 6.1 | 4.5[c] | 8.5 | +0.2[c] |
| Suboxic sediment | 375[d] | 87.4 | 15.3 ± 10.6[b] | 29 | −0.30 |
| Oxic sediment | 26.1[d] | 6.5 | 22.9[c] | 43 | −0.395 |
| Shelf sediment[e] | 431 ± 175 | 100 ± 41 | 57 ± 22 | 108 ± 42 | – |
| Shelf sediment[f] | 685 [734, 1830] | 160 [171, 427] | 184 [154, 252] | 346 [291, 476] | – |
| Updated Suboxic sediment[g] | 806 ± 175 | 188 ± 41 [g] | 72 ± 25 [g] | 136 ± 47 [g] | – |

[a]Ref. 21.
[b]Flux and uncertainty in ref. 20.
[c]The output in oxic sediment includes marine carbonates, Mn-oxides, oceanic crust alteration, pelagic clays, and coastal retention[5].
[d]Ref. 3.
[e]Estimated with the correlation between sedimentary reductive removal and DO consumption (this study) and the sediment oxygen consumption rates[53]; the uncertainties (1 SD) were propagated from the correlation between sedimentary reductive removal of Re and U and sediment oxygen consumption (this study) and the sediment oxygen consumption rate for the global shelf sediments[53].
[f]Estimated by extrapolating the compiled fluxes in shelf sediments to the global suboxic sediments, reported as median [lower, upper bounds] of 95% confidence interval;
[g]Sum of the sinks in shelf sediment (this study) and the suboxic sediment beyond shelf reported in refs. 3,20. The uncertainties were propagated from those associated with these two terms.

River estuary (southeast China), respectively[62,63]. Interestingly, it was identified that Re could be buried at a rate ~4-fold higher than oxic marine sediments via clay adsorption and biological uptake in an estuary-like lagoon[17], which could be an additional sink within the shelf area. Sorption onto Fe-oxyhydroxides and colloidal flocculation during estuarine transport and/or encompassing coastal blue carbon ecosystems was supposed to be a "coastal retention" sink of U[20]. However, recent studies suggest U addition in estuarine and blue

carbon systems, although the accurate fluxes are not well-constrained[64,65]. Therefore, the addition/removal of Re and U during transport across estuaries and coastal blue carbon systems may bias their fluxes to the ocean[8,20]. As high-precision analytical techniques have progressed, it is important to revisit the contribution of Re and U from these systems.

Moreover, glacial-interglacial cycles of the size of shelf sediment sinks may also contribute to the oceanic budgets considering the long

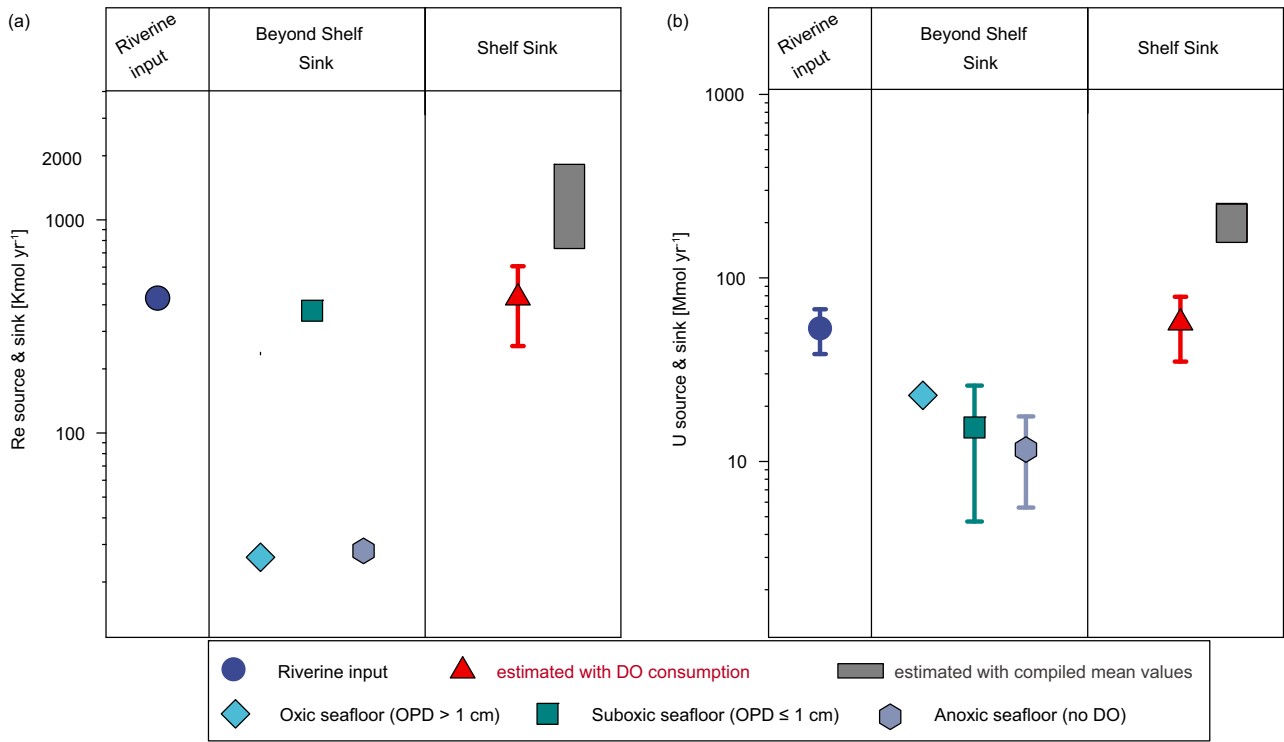

**Fig. 6 | Sources and sinks of Re and U in the modern ocean.** The terms of Re (**a**) and U (**b**) are shown with the updated shelf sediment sink[3,5,8,20,21]. The definitions of the oxic, suboxic, and anoxic settings are the same as those in Fig. 5. The error bars of the shelf sinks represent the uncertainties (1 SD) propagated from the correlations of sedimentary reductive removal of Re and U with sediment oxygen consumption and the sediment oxygen consumption rates for the global shelf sediments[53]. The error bars in other terms (riverine input and beyond shelf sinks) denote the uncertainties (1 SD) initially reported in refs. 5,8,20.

ocean residence times of Re and U. The ~50% decrease of total shelf area upon seal level lowering during glacial periods could result in smaller sinks over the global shelf. Furthermore, during the glacial periods, the decreased organic matter supply and increased oxygenation in the upper ocean, including the continental shelves impinging the modern oxygen minimum zone[7], result in declines of oxidant (DO, metal oxides) requirement for OC decomposition within the sediments. Therefore, lower sedimentary reductive sinks of Re and U over global shelves during cold periods are expected.

Including the shelf sediment sink is remarkable for the oceanic $\delta^{238}U$ budget. Recent studies suggest a systematically higher oceanic $\delta^{238}U$ sink than the bulk continental crust[2,8] and ascribe it to the non-steady state of modern oceanic U budget, too high $\delta^{238}U$ of the net oceanic sink, and/or a higher riverine $\delta^{238}U$ than that of bulk continental crust. Our estimate implies that the modern oceanic U budget might indeed be imbalanced. Additionally, the importance of shelf sediment sinks may alter the modern oceanic $\delta^{238}U$ budget. For example, combining the updated shelf sediment sink (this study) with the same $\delta^{238}U$ of the suboxic sink (-0.30‰; ref. 8) results in a total oceanic $\delta^{238}U$ sink of -0.30‰, which is closer to the riverine influx of -0.34‰ than previous estimates (-0.26‰; refs. 2,8). Thus, this shelf sediment sink helps to close the oceanic $\delta^{238}U$ budget.

## Methods
### Sample collection
Samples were collected on the East China Sea shelf during summer (Aug 2017 and July–Aug 2021; labeled with "-S") and winter (Dec 2018; labeled with "-W") (Supplementary Fig. 1). Sediments were collected using a box corer ($20 \times 20 \times 60$ cm$^3$) and checked visually undisturbed. Subcores were taken by inserting transparent PVC tubes (inner diameter: 65 mm) into bulk sediments. Subcores for DO measurement were temporarily stored with an open top and bubbled with an $N_2/CO_2$ mixture to keep the DO concentrations at the bottom water levels.

Porewater samples were extracted from sediment subcores using Rhizon samplers and pre-cleaned 10-mL PE syringes[66]. Aliquots for trace metal analysis were acidified to pH = 1.6 with Optima grade $HNO_3$, while aliquots for nutrients and anions (Cl$^-$ and $SO_4^{2-}$) were stored at −20 °C and 4 °C until analysis. Seawater samples were collected using Niskin bottles mounted on a CTD rosette. These samples include the bottom water (~2 m above the seafloor) collected during the winter of 2018, as well as the vertical profiles of the sites sampled in the summer of 2021 and at a distal site (C18: 30.08°N; 124.49°E) in the summer of 2019. They were filtered with Polyethersulfone membrane (pore size: 0.22 μm), subsequently acidified and stored in the same manner as porewater.

### Analysis of trace metals
Concentrations of Re were measured by Q-ICP-MS (iCAP RQ; Thermo Fisher Scientific) after preconcentration and purification of 5–10 mL porewater/seawater samples using isotope dilution method with anion resin (1 mL AG1-X8 resin, 200–400 mesh). Sample solutions were spiked with $^{185}$Re-enriched solution and equilibrate for 24 h. Samples were then processed through anion resin columns conditioned with 1 M HCl solution. The Re fraction was collected with 10 mL 8 M $HNO_3$ after washing the sample matrix with 10 mL 1 M HCl solution. The Re fraction was evaporated and re-dissolved in 4 mL 2% $HNO_3$ for instrumental analysis. Iron, manganese, and uranium concentrations were diluted 20 - 100-fold and determined using ICP-MS with He as a collision to minimize polyatomic interferences. A matrix-matched external calibration with NaCl was applied using Rh as the internal standard. Certified reference materials NASS-7 and SLRS-6 (for Fe, Mn, Re, and U) were used for quality control. The influence of the porewater/seawater matrix on the accuracy of Fe, Mn, and U measurements was assessed using a spiked seawater standard. The results are consistent within 10% of the certified or previously reported values (Supplementary Table 5). For Re, replicate aliquots of surface seawater from the ECS were

measured to be $38.0 \pm 0.4$ pmol $L^{-1}$ ($N = 8$). The external reproducibility of replicate analyses was better than 2% for Re and 5% for Mn, Fe, and U.

## Analysis of ancillary parameters

The activities of $^{224}Ra$ and $^{228}Th$ in bulk sediments and porewater were determined with the RaDeCC systems[67]. Briefly, the sediment subcores were sliced into 1-cm thick slabs. The slabs were slurried with MilliQ water in Teflon beakers. After adjusting the pH of the slurry to 8–9, $KMnO_4$ ($8.0\,g\,L^{-1}$) and $MnCl_2$ ($3.0\,g\,MnCl_2 \cdot 4H_2O\,L^{-1}$) solutions were added to form a suspension of $MnO_2$. The samples were filtered onto a GFF filter (diameter: 142 mm; pore size: 0.7 μm). The filter was initially counted on a RaDeCC system for ~6 h and re-counted after ~10 and ~25 days, respectively. The activities of $^{224}Ra$ and $^{228}Th$ in bulk sediment were calculated from the first and second/third measurements. Porewater $^{224}Ra$ was pretreated and measured in the same fashion but skipped the addition of MilliQ water.

Ammonium and nitrate were determined using a flow injection analyzer (Tri-223 Auto-Analyzer) and a 4-channel segmented flow auto-analyser (Bran-Lube AA3). DO in the water column was measured using the Winkler titration method. Fine-scale DO profiles (resolution: 0.3 mm) in sediment porewater were determined using a Unisense or custom-made gold amalgam microelectrode[68]. Anions ($SO_4^{2-}$ and $Cl^-$) were determined after a 200-fold dilution by ion chromatography (Metrohm AG) using IAPSO seawater for calibration and matrix correction. The analytical uncertainties for an in-house seawater sample were <1%. Porosity was estimated with the weight difference of the sediment after drying to a constant weight at 60 °C.

## Calculation of the Re and U fluxes based on the $^{224}Ra/^{228}Th$ disequilibria

Sedimentary reductive removal can create $[Re]_{diss}$ and $[U]_{diss}$ gradients in sediment porewaters, which are traditionally used to quantify the diffusive flux following Fick's first law[69]: $F_M = \varphi \times D_S^i \times \partial C/\partial z$. Here, $\varphi$ is the porosity, $D_S^i$ is the effective diffusion coefficient in sediments calculated using the molecular diffusion coefficient in seawater ($D^i$) and corrected for tortuosity $D_S^i = D^i/(1 - 2 \times \ln\varphi)$, and $\partial C/\partial z$ represents the concentration gradient of dissolved components ($^{224}Ra$, Re, U; Supplementary Table 6). Irrigation dominates solute exchange between porewater and bottom water in shelf sediments[14,31,36–38]. Thus, the reductive removal fluxes of Re and U ($F_i$, positive values denote removal) can be estimated with the $^{224}Ra/^{228}Th$ disequilibria approach, which incorporates the flux enhanced by irrigation[37]:

$$F_i = -F_{Ra} \cdot \left( D_S^i / D_S^{Ra} \right) \cdot \left( \frac{\partial C^i/\partial z}{\partial C^{Ra}/\partial z} \right) \quad (1)$$

where $F_{Ra}$ denotes the flux of total $^{224}Ra$ (dpm $m^{-2}$ $d^{-1}$) and is calculated as the product of the decay constant of $^{224}Ra$ ($\lambda_{Ra} = 0.189$ $d^{-1}$) and the integration of the deficit of total $^{224}Ra$ ($A_{^{224}Ra}$) relative to $^{228}Th$ ($A_{^{228}Th}$): $F_{Ra} = \lambda_{Ra} \cdot \int_0^z \left( A_{^{228}Th} - A_{^{224}Ra} \right) dz$[67]. The deficit ($^{224}Ra/^{228}Th$ ratio <1) of the solids was confined within the depth of ~10 cm, although they could be scattered due to inhomogeneous sediment particle mixing (Fig. 1c). We here integrate the upper 10 cm to derive the total $^{224}Ra$ depletion fluxes. Amplification factor ($\xi$) is then defined as[31]: $\xi = \frac{F_{Ra}}{\varphi \times D_S^{Ra} \times \partial C^{Ra}/\partial z}$.

The calculated amplification factor represents the mean state of irrigation within the upper few cm where $^{224}Ra/^{228}Th$ disequilibria occur. The fluxes and concentration gradients of $^{224}Ra$ for the 2021 cruise, together with the previously reported dataset for the 2017 and 2018 cruises (ref. 31), are presented in Table 1 and Supplementary Table 6. We determined the Re and U concentration gradients by modeling a linear fit from the surface layer to the depth where significant gradient change occurs (Supplementary Fig. 7) to account for the influence of

irrigation indicated by the $^{224}Ra/^{228}Th$ disequilibria. This allows us to estimate the reductive removal fluxes of Re and U. For the sites characterized with complicated removal-release in the upper sediments indicated by distinct zig-zag patterns within the upper few cm (Supplementary Fig. 7), their concentration gradients and the resulting removal fluxes were not calculated due to the large uncertainties. The uncertainties associated with the site-specific reductive removal fluxes of Re and U were propagated from the errors of the $^{224}Ra$ fluxes and the concentration gradients of dissolved $^{224}Ra$, Re, and U.

For the sites with Re and U flux out of the sediment–water interface ($F_{out}$), their $F_{out}$ were also estimated using Eq. (1). The concentration gradient at the sediment–water interface is calculated from the first porewater (at 1 cm) and the seawater concentrations (assumed to be equal to those of the overlying water column). When the first porewater concentrations of Re and U are lower than the water column, $F_{out}$ is set to zero.

## Data availability

The data generated in this study are provided in the Supplementary Information and are also available in the Figshare database under accession code: https://doi.org/10.6084/m9.figshare.24807534.

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

## Acknowledgements

This work was supported by the National Natural Science Foundation of China (Grant No. 92358301, 92058205, 42106037), the Strategic Priority Research Program of Chinese Academy of Sciences (XDB40010200), and the "GeoX" Interdisciplinary Research Funds for the Frontiers Science Center for Critical Earth Material Cycling, Nanjing University. We thank Wei Liu, and Tong Wu for their kind help during sample collection and analysis.

## Author contributions

T.C., Q.H. and P.C. designed the study. Q.H., Y.C., Y.Q., L.W. and Y.L. collected and analyzed the samples. J.G. contributes to the development of Re concentration analysis using isotope dilution. Q.H. and T.C. made the interpretations and wrote the manuscript with inputs from all authors.

## Competing interests

The authors declare no competing interest.
