## [Peer Review File · Nature Communications]

Overlooked shelf sediment reductive sinks of dissolved rhenium and uranium in the modern oceanReviewer #1 (Remarks to the Author):

Hong et al. present the Re and U concentrations in the pore water of shelf sediments from the East China Sea. The benthic fluxes of Re and U were estimated using the $^{224}\text{Ra}/^{228}\text{Th}$ disequilibrium method to account for the influence of sediment transformation/irrigation. Although the data presented are a valuable contribution to a better understanding of the biogeochemistry of Re and U in the marine environment, I have significant problems with the authors' interpretation of the data. Unfortunately, I cannot recommend the publication of the manuscript in its current form. But before I go into the interpretation, I have a question about Supplementary Figures 1 and 2. The sampling in summer 2017 is distinguished from the sampling in summer 2021 by red bold circles. However, Supplementary Figure 2 shows that sampling at stations F4 and F3 was also carried out in summer 2017 (the text on sampling – lines 118 to 126 in main manuscript text – does not help to clarify this issue). In addition, the Supplementary Figure 2 shows the station designated as B7, which does not exist on the map in Supplementary Figure 1. The same applies to the sampling station marked as HN (Supplementary Figure 2 compared to Supplementary Figure 1). The same applies to the sampling stations for uranium.

Looking at the Re pore water depth profiles, we can see that the Re concentration in the upper 1 to 2 cm sediment depth is elevated in 11 out of 17 cores (one more or less) compared to the Re concentration in seawater (accidentally, the Re concentration profiles of the East China Sea are not shown anywhere – at least at one point, while the bottom water Re concentration are shown only in a few cases), if we assume that the dissolved Re concentration in the East China Sea is in the range of reported values. A similar phenomenon has already been observed by Morford et al. (Ref. 1 and 2). In Ref. 2, Morford et al. “the flux into the sediments” and the “flux out of the sediments” for a particular site is also depicted. I don't think that the observed feature can be called as benthic uptake (e.g. line 137 in the main text), because in most cases the authors have the flow out of the sediment across the sediment-water interface and the Re-porewater maxima below which pronounced Re-uptake occurs. I think that the authors have only tried to estimate the Re uptake occurring below sediment-water interface, while the obvious Re flux from the sediment observed in most pore water profiles is neglected.

I think the authors are commenting on the Re peak in subsurface pore water observed in the text at station B7. But the same is true for the majority of the data, except that the Re pore water peak is near the sediments surface. Also, next time I would suggest the authors try to do a denser pore water profile for the top 5 cm of the sediments, as Morford et al. (Ref. 1 and 2).

It is also unclear that the red line representing the linear regression sometimes reaches the sediment-water boundary and sometimes goes beyond it, although the Re-soil water concentration is not given for these sites. It is therefore not clear which Re-boundary concentrations have been included in the calculations.

I would also like to draw the authors' attention to Ref. 3.

References

1. Morford et al., A model for uranium, rhenium, and molybdenum diagenesis in marine sediments based on results from coastal locations. *Geochimica et Cosmochimica Acta* 73 (2009) 2938-2960.
2. Morford et al., Rhenium geochemical cycling: Insights from continental margins. *Chemical Geology* 324-325 (2012) 73-86.
3. Danish et al. Non-conservative removal of dissolved rhenium from a coastal lagoon: Clay adsorption versus biological uptake. *Chemical Geology* 580 (2021) 120378.

Reviewer #2 (Remarks to the Author):

A lot of different types of carefully executed measurements were made on these cruises. Some of the method details are already published (and so are referenced in the short methods section). As a whole, I found the paper easy to read and attention grabbing. While this isn't the first estimate ever of shelf sinks, the paper combines the literature data with new ECS measurements to look at the 'shallow shelf' question at a global scale. I agree that 'A key advantage of this $^{224}\text{Ra}/^{228}\text{Th}$ disequilibrium method over traditional sediment incubation and modeling methods is that it captures all physical processes that affect solute transfer across the sediment-water interface but does not alter the system' and that these radioisotope-based estimates are a unique application here to looking at budgets. Some of the data is reused from earlier studies, but the authors utilize

interesting applications of existing methods/techniques in a way that provides a unique contribution to the field. I thought the authors did a good job of 'considering the options' starting at line 259 ('Shelf sediments as an overlooked sink'). Budget parameters were often determined in 2 ways to support the conclusions. Implications and 'what ifs' were discussed, which shows that the authors have both considered potential issues with higher sinks and, if real, what higher sinks mean for this field and beyond (i.e. paleo-interpretations, etc.). If the larger sink for U is real and the system is not in balance for U, this is very important for the oceanographic community to know.

I have a few broad comments/concerns that would be helpful for the authors to clarify (whether in responses or in the manuscript text itself). In my opinion, addressing these questions/areas would help to strengthen the conclusions and assuage doubts.

1. Potential variability of the datasets used (temporal questions) and how representative the dataset is of global shelf conditions for depths <200 m.

One thing that I would want to see (without looking at references) is what is the timescale of the summer and winter sampling in Figure 1 and Table 1? Could this go in the caption of Fig 1? Essentially, were all the samples from summer taken the same day? Or the same week? The reason this is of interest is to see how variable the Ra/Th could be (averaging over 10 days) from week to week or month to month on the shelf. Would we see much much higher or much much lower a few weeks later? In short, are these representative conditions on this shelf throughout most of the summer and winter? How does the '10 day average' for these radioisotope ratios change at one location over the entire summer? Are there temporal studies in the literature that can be cited here to indicate this range? Two summer dates are shown, although they aren't from the same locations (Fs are down at the bottom of the map and E/Y are in the middle). The results do seem comparable on the scale in Fig. 2. Since there is so much variability, as noted in lines 166-167 (when comparing to other studies) and Fig 1/Fig 5, it would be helpful to know how changeable the Ra/Th ratios are.

2. Potential variability of the datasets used (spatial questions) and how representative the dataset is of global shelf conditions for depths <200 m.

Lines 63-68: 'The contrasting conditions...the global oceanic Re and U budgets.' I agree that it's a good region to look at different patterns because of summer/winter changes, my question would be do most shelves globally look like this one and have similar processes? Do we know or have an idea of how representative they might be? When the estimates are extrapolated to the global model, how does this shelf compare to the typical conditions of the 'average' shelf?

If I look at Fig 3 and Fig 2, two of the really high flux points are northern winter points. Do we know (I assume yes) about the shelf dynamics here and can we attribute these patterns to summer vs. winter and not north shelf vs. south shelf spatial trends? Is there more upwelling in one location vs. other? Are the sediment types similar? Are the regressions in Fig 3. still strong (>0.5) if the two northern points are taken away? I realize these measurements are very difficult and occupying the same locations twice might not have been possible. There is some overlap around 31 in summer and winter points. A short comment/discussion from the authors on how the patterns/trends are clearly related to season and not spatial variability would be welcome (especially for those that don't know the physics and sedimentary systems of this area well).

Line 286 (for U) and early for Re use the median flux extrapolated over the are of global suboxic sediments. How realistic is the resulting estimate? Are all shelves expected to be like the ECS or said in another way, if we know shelves are dynamic/different what are the bounds on these estimates? I agree that the locations in Fig 4 are covering some of the ocean's broadest shelves, however, this doesn't mean that these conditions will be everywhere. A lot of the Hudson Bay numbers look very low compared to the ECS. In short, how homogenous can we expect the shelves to be (i.e. if there is a mix of types, how many are going to have the 'interesting' summer/winter conditions seen in the ECS)? Is there a way to characterize the shelves globally with different types (broad categories) instead of just applying an average or median of the available data to the entire shelf area globally?

Along these same lines, how are the 'uncertainties' calculated, or errors extrapolated? I see data from this study (with errors) and data from the literature (no errors) in Supp. Table 4. These were the data used to get the budget estimates starting at Line 259. Medians were reported in this section, but ranges and standard deviations might be helpful too. This could go in Table 2. I'm not sure what the +/- in this table mean (i.e. what do they represent?).

Additional Line by Line Comments:

From Line 9, it sounds like the authors (from the literature) didn't anticipate shelves being a sink for Re previously, and so that with this study there is now an extra sink for Re (i.e. if the shelf sink is comparable to suboxic/anoxic sinks that were found previously then the total sink has now increased...because otherwise things would still be the same). Just clarifying here that this is what the authors intended: 'Our extrapolation suggests the shelf sinks are comparable to (for Re) or higher than (~4-fold for U) previous estimations of the suboxic/anoxic sinks. These results suggest that the modern budget of Re and U may be imbalanced and/or their sources are substantially underestimated.'

Line 38: 'In fact, the established sinks were based on limited data from continental margins of water depths >300 m, 8.' Based on a quick read of the text, I am assuming that source 8 (Dunk et al, 2022) uses the location in Table 1 to make the shelf sink estimates. I see at least two locations where water depths are <300 m, which would contradict what is said in lines 38-40 of the manuscript. I believe Saanich Inlet is maximum 230-250m and most depths from the Walvis Bay study (Veeh et al) are 119-225m. It would be helpful if the authors could clarify this point regarding the depth range cutoff used. If Table 1 is the source of the estimates being referred to in source 9, then there seems to be a range in settings and depths (vs. a limited one). I could be mistaken how Table 1 is used, but since this is an important point to the significance of this study (i.e. showing that shallow sediments could be greater than assumed), this part of the text about previous estimates should be very clear.

Lines 153, 156 and later uses of 'amplification': Please explain what is meant here by 'extreme amplification of sediment area' and define this term - I had to go to source 18 to get information about the anthropogenic influences. Since amplification is also mentioned in a different context in Line 188, the term should be defined to account for a wider audience reading Nature Communications. My question here is also, do we anticipate the amplification factor due to anthropogenic disturbance to be common? If this is not going to happen in most locations on most shelves, then omitting is fine. The authors don't need to add a huge amount of detail, just indicating how anomalous this is expected to be would help rule this out for use in any extrapolation.

Fig 1: Overall it's a bit small to read but I like the setup here and it helps the reader quickly understand the Ra/Th and Re, U data. The grey bar is helpful, as are the caption details.

Fig 2: Good visual. Liked it.

Fig 3: Some overlap of text with the points on (e) but overall this was a readable figure. Well structured.

Fig 4: Good figure. If you needed to cut anything, (a) could be removed. What I was hoping to see is the shelf variability in b/c/d (and that was shown). You can see the shelf depths much more clearly in b/c/d and so those panels would be enough for me with a caption indicating (as is already done) where each is from.

Figure 5: Overall this figure is a good representation of the data and is important. However, the sideways text is a bit confusing/messy and I would recommend using a bracket of some kind at the top or bottom of the plots to indicate data from this study (left, use bracket to cover the red/blue) and the data from the referenced sideways text studies (red/green/purple).

Figure 6: Same comment as fig. 5. The figure will look more professional without the

squished/sideways text.

Reviewer #3 (Remarks to the Author):

The importance of shallow shelf sediments on the mass balance for U and Re requires more data and closer study, as both are used for reconstructing past anoxic conditions. U has been used more than Re for this purpose, but this is primarily due to the paucity of available Re data. The authors further our quantitative understanding of the importance of irrigation through using $^{224}\text{Ra}/^{228}\text{Th}$ disequilibria, which is particularly important in these shallow shelf sediments. Overall, I would like to see this manuscript published. We use U extensively (both U concentration in sediments and U isotopes in carbonates), which requires understanding sinks and sources, while Re has the potential to be an extremely valuable tracer, but only if more data is obtained from diverse marine locations. The unexpected greater removal in winter compared to summer is also a surprising twist that will encourage the community to consider the seasonal changes in these shallower sediments in different ways. I do have a few minor comments that the authors should consider.

- Lines 45-46: The authors suggest that Re and U concentration and isotopic composition are 'emerging proxies', but that statement suggests a recentness that may not be appropriate. U concentrations and isotopic reconstruction of past conditions have been used more extensively than indicated in refs 2-5, and it is for exactly this reason that a better understanding of sinks/sources is required.
- Lines 59-60: the authors cite a lack of Re isotope data, but in 2020 Dickson et al started this process (The rhenium isotope composition of Atlantic Ocean seawater, GCA 287), along with Dellinger et al. in 2021 (Fractionation of rhenium isotopes in the Mackenzie River basin during oxidative weathering, EPSL, 573).
- Lines 91-93: The authors suggest that authigenic accumulation rates are not used for Re and U due to the high detrital contribution that could hinder the accuracy of the accumulation rate. But is this the case for Re, which has an astonishingly small detrital concentration? More support for this statement would be appropriate.
- Figure 1: I found it extremely difficult to see the pore water profiles due to the volume of data presented on this figure. I ask whether this might be two figures with some of the profiles separated for each season so that they can be more easily seen, or whether the authors want to consider averaging some of the data and presenting a standard deviation for that profile.
- Line 133: The authors cite the possibility of colloidal phases to explain incomplete removal, but other ideas instead center around a minimum U concentration required for biological mediation of its removal which should be considered (Lovely et al., 1991, Nature, 350).

Minor comments:

- Line 73: Rather than 'inspires' I would suggest 'causes'.
- Line 166: The decay constant for ^{224}Ra didn't appear in my version of equation 1, so it was unclear if there was a formatting issue.
- Line 342: Should be 'lowering' instead of 'lowing'
- Table 2: The percentage symbols can be removed from within the table since they are in the column heading.

Response to Reviewers' comments

We thank the reviewers and editor for the detailed comments on our manuscript. We have thoroughly revised the manuscript and provided additional data on Re and U concentrations in seawater and bulk sediments based on the comments. Please see our point-by-point response below.

Reviewer #1 (Remarks to the Author):

Hong et al. present the Re and U concentrations in the pore water of shelf sediments from the East China Sea. The benthic fluxes of Re and U were estimated using the $^{224}\text{Ra}/^{228}\text{Th}$ disequilibrium method to account for the influence of sediment transformation/irrigation. Although the data presented are a valuable contribution to a better understanding of the biogeochemistry of Re and U in the marine environment, I have significant problems with the authors' interpretation of the data. Unfortunately, I cannot recommend the publication of the manuscript in its current form.

Reply 1: We thank the reviewer for valuing the contribution of our study. We have carefully considered all the comments in our revised manuscript and replied below.

But before I go into the interpretation, I have a question about Supplementary Figures 1 and 2. The Sampling in summer 2017 is distinguished from the Sampling in summer 2021 by red bold circles. However, Supplementary Figure 2 shows that Sampling at stations F4 and F3 was also carried out in summer 2017 (the text on Sampling – lines 118 to 126 in main manuscript text – does not help to clarify this issue). In addition, the Supplementary Figure 2 shows the station designated as B7, which does not exist on the map in Supplementary Figure 1. The same applies to the sampling station marked as HN (Supplementary Figure 2 compared to Supplementary Figure 1). The same applies to the sampling stations for uranium.

Reply 2: We apologize for the inconsistency of the station names. The stations sampled in the summer of 2017 were initially named as those presented in the original Supplementary Figure 2), but all were re-named in previous papers (Shi et al., 2019; Wei

et al., 2022). We have now corrected them for consistency (Supplementary Fig. S7). The sampling was only performed once at all stations.

References:

Shi et al. Large benthic fluxes of dissolved iron in China coastal seas revealed by $^{224}\text{Ra}/^{228}\text{Th}$ disequilibria. *Geochimica Cosmochimica Acta* 260 (2019), 49-61.

Wei et al. Winter mixing accelerates decomposition of sedimentary organic carbon in seasonally hypoxic coastal seas. *Geochimica Cosmochimica Acta* 317 (2022), 457-471.

Looking at the Re pore water depth profiles, we can see that the Re concentration in the upper 1 to 2 cm sediment depth is elevated in 11 out of 17 cores (one more or less) compared to the Re concentration in seawater (accidentally, the Re concentration profiles of the East China Sea are not shown anywhere – at least at one point, while the bottom water Re concentration are shown only in a few cases), if we assume that the dissolved Re concentration in the East China Sea is in the range of reported values. A similar phenomenon has already been observed by Morford et al. (Ref. 1 and 2). In Ref. 2, Morford et al. "the flux into the sediments" and the "flux out of the sediments" for a particular site is also depicted. I don't think that the observed feature can be called as benthic uptake (e.g. line 137 in the main text), because in most cases the authors have the flow out of the sediment across the sediment-water interface and the Re-porewater maxima below which pronounced Re-uptake occurs. I think that the authors have only tried to estimate the Re uptake occurring below sediment-water interface, while the obvious Re flux from the sediment observed in most pore water profiles is neglected.

Reply 3: We thank the reviewer for this important concept-changing comment. We agree that this is an issue that we have neglected in the original manuscript. We have now added a section entitled "Sources of Re and U for sedimentary reductive removal" to discuss the processes that may be responsible for the elevated Re and U concentration in porewater. We then clarified that we are quantifying the sedimentary reductive removal

occurring below the sediment-water interface (instead of “benthic uptake from bottom water”) that ultimately results in the enrichment of Re and U in the modern shelf sediments.

“The elevated porewater Re and U concentrations at the sediment–water interface of some sites exceed those found in the seawater (Fig. 1a,b), indicating additional sources supplying porewater Re and U for reductive removal, apart from irrigation and diffusion of dissolved species from bottom seawater (Supplementary Fig. S5)^{12, 13, 14, 15, 18}. Under oxic bottom water, enhanced bioturbation may result in the re-oxidation of reduced Re and U^{14, 18}. However, the most distinct enrichments of core-top porewater $[Re]_{diss}$ and $[U]_{diss}$ are observed in summer with lower bioturbation compared to winter (Fig. 1). In this regard, re-oxidation of reduced Re and U is unlikely to be the dominant source for the observed pattern at these sites. Instead, the decomposition and transformation from freshly deposited particles (particularly organic matter) at the sediment–water interface might have released relatively large amounts of Re and U compared to the reductive removal^{14, 16, 18, 19}. The “particulate source” scavenged from seawater could even have exceeded the reductive removal flux and resulted in net dissolved flux back to the water column (Supplementary Fig. S5).

Moreover, the broad maxima of $[Re]_{diss}$ and $[U]_{diss}$ at St. B3-W, E4-S, F1-S, and Y7-S could not be supported merely by their release at the sediment–water interface³⁰. Alternatively, they may be indicative of enhanced Re and U release from the solid phases into porewater at these depth intervals, probably due to intense sediment reworking. Below these depths, Re and U are steadily removed to reach the typical asymptotic minima.

Regardless of the routes (dissolved versus particulate) that supply Re and U to the porewaters, it is the reductive removal that ultimately results in the enrichment of Re and U in the modern shelf sediments. Our data on Re and U concentrations in the corresponding solid phases provides further support for the reductive enrichment process (Supplementary Fig. S4). The $^{224}Ra/^{228}Th$ disequilibria approach, which incorporates the influence of irrigation in regulating solute fluxes, is used to estimate the reductive

removal fluxes of Re and U (F_i , positive values denote removal)³⁶. A detailed description of the calculation is provided in Methods.”

See the above text in Line 166-195.

We have also added a section of “Calculation of sedimentary reductive removal fluxes of Re and U from the ²²⁴Ra/²²⁸Th disequilibria” to describe the details for the flux estimation:

“Sedimentary reductive removal can create $[Re]_{diss}$ and $[U]_{diss}$ gradients in sediment porewaters, which are traditionally used to quantify the diffusive flux following Fick's first law⁶⁶: $F_M = \phi \times D_s^i \times \partial C / \partial z$. Here, ϕ is the porosity, D_s^i is the effective diffusion coefficient in sediments calculated using the molecular diffusion coefficient in seawater (D^i) and corrected for tortuosity $D_s^i = D^i / (1 - 2 \times \ln \phi)$, and $\partial C / \partial z$ represents the concentration gradient of dissolved components (²²⁴Ra, Re, U; Supplementary Table S6). Irrigation dominates solute exchange between porewater and bottom water in shelf sediments^{14, 30, 35, 36, 37}. Thus, the reductive removal fluxes of Re and U (F_i , positive values denote removal) can be estimated with the ²²⁴Ra/²²⁸Th disequilibria approach, which incorporates the flux enhanced by irrigation³⁶:

$$F_i = -F_{Ra} \left(\frac{D_s^i}{D_s^{Ra}} \right) \left(\frac{\partial C^i / \partial z}{\partial C^{Ra} / \partial z} \right) \quad (1)$$

where F_{Ra} denotes the flux of total ²²⁴Ra ($dpm\ m^{-2}\ d^{-1}$) and is calculated as the product of the decay constant of ²²⁴Ra ($\lambda_{Ra} = 0.189\ d^{-1}$) and the integration of the deficit of total ²²⁴Ra

(A_{224Ra}) relative to ²²⁸Th (A_{228Th}): $F_{Ra} = \lambda_{Ra} \int_0^z (A_{228Th} - A_{224Ra}) dz$ ⁶⁴. The deficit

(²²⁴Ra/²²⁸Th ratio <1) of the solids was confined within the depth of ~10 cm, although they could be scattered due to inhomogeneous sediment particle mixing (Fig. 1c). We here integrate the upper 10 cm to derive the total ²²⁴Ra depletion fluxes. Amplification

factor (ξ) is then defined as³⁰: $\xi = \frac{F_{Ra}}{\phi \times D_s^{Ra} \times \partial C^{Ra} / \partial z}$. The calculated amplification factor

represents the mean state of irrigation within the upper few cm where ²²⁴Ra/²²⁸Th

disequilibria occur. The fluxes and concentration gradients of ^{224}Ra for the 2021 cruise, together with the previously reported dataset for the 2017 and 2018 cruises (ref.³⁰), are presented in Table 1 and Supplementary Table S6. We determined the Re and U concentration gradients by modeling a linear fit from the surface layer to the depth where significant gradient change occurs (Supplementary Fig. S7) to account for the influence of irrigation indicated by the $^{224}\text{Ra}/^{228}\text{Th}$ disequilibria. This allows us to estimate the reductive removal fluxes of Re and U. For the sites characterized with complicated removal-release in the upper sediments indicated by distinct zig-zag patterns within the upper few cm (Supplementary Fig. S7), their concentration gradients and the resulting removal fluxes were not calculated due to the large uncertainties. The uncertainties associated with the site-specific reductive removal fluxes of Re and U were propagated from the errors of the ^{224}Ra fluxes and the concentration gradients of dissolved ^{224}Ra , Re, and U.” See the text in Line 443-476.

We have now added vertical distribution of Re and U in the water column of some stations and found relatively homogeneous concentrations except for one elevated point at St. E1-W near a refuge harbor (Supplementary Fig. S3).

I think the authors are commenting on the Re peak in subsurface pore water observed in the text at station B7. But the same is true for the majority of the data, except that the Re pore water peak is near the sediments surface. Also, next time I would suggest the authors try to do a denser pore water profile for the top 5 cm of the sediments, as Morford et al. (Ref. 1 and 2).

Reply 4: We have neglected to discuss the subsurface peaks at other stations and the zig-zag patterns at some sites (St. B3-W, E4-S, F1-S, and Y7-S) in the original manuscript. Please see our detailed reply above.

We appreciate the reviewer’s suggestion and will perform high-resolution sampling in future work.

It is also unclear that the red Line representing the linear regression sometimes reaches the sediment-water boundary and sometimes goes beyond it, although the Re-soil water

concentration is not given for these sites. It is therefore not clear which Re-boundary concentrations have been included in the calculations.

Reply 5: During revision, we have refined how we calculate the sedimentary reductive flux, see Supplementary Fig. S7. The regression (denoted by red lines) was done with the data below the sediment-water interface (filled circles). For some sites characterized with distinct zig-zag patterns, their concentration gradients were not calculated due to the large uncertainties. A detailed description of the linear regression was added in the Method (Line 466-473):

"We determined the Re and U concentration gradients by modeling a linear fit from the surface layer to the depth where significant gradient change occurs (Supplementary Fig. S7) to account for the influence of irrigation indicated by the $^{224}\text{Ra}/^{228}\text{Th}$ disequilibria. This allows us to estimate the reductive removal fluxes of Re and U. For the sites characterized with complicated removal-release in the upper sediments indicated by distinct zig-zag patterns within the upper few cm (Supplementary Fig. S7), their concentration gradients and the resulting removal fluxes were not calculated due to the large uncertainties."

I would also like to draw the authors' attention to Ref. 3.

Reply 6: We appreciate the reviewer for pointing out this work. It supports our interpretation of Re scavenging from the water column by biological organic matter, which serves as another removal mechanism of Re into sediments. The study also documented an unaccounted Re sink in the coastal lagoon that is ~4 times higher than that in oxic marine sediments. We have added the information in the revised manuscript (Line 54, 63-65 and Line 367-370).

References

1. Morford et al., A model for uranium, rhenium, and molybdenum diagenesis in marine sediments based on results from coastal locations. *Geochimica et Cosmochimica Acta* 73 (2009) 2938-2960.
2. Morford et al., Rhenium geochemical cycling: Insights from continental margins. *Chemical Geology* 324-325 (2012) 73-86.

3. Danish et al. Non-conservative removal of dissolved rhenium from a coastal lagoon: Clay adsorption versus biological uptake. *Chemical Geology* 580 (2021) 120378.

Reviewer #2 (Remarks to the Author):

A lot of different types of carefully executed measurements were made on these cruises. Some of the method details are already published (and so are referenced in the short methods section). As a whole, I found the paper easy to read and attention grabbing. While this isn't the first estimate ever of shelf sinks, the paper combines the literature data with new ECS measurements to look at the 'shallow shelf' question at a global scale. I agree that 'A key advantage of this $^{224}\text{Ra}/^{228}\text{Th}$ disequilibrium method over traditional sediment incubation and modeling methods is that it captures all physical processes that affect solute transfer across the sediment–water interface but does not alter the system' and that these radioisotope-based estimates are a unique application here to looking at budgets. Some of the data is reused from earlier studies, but the authors utilize interesting applications of existing methods/techniques in a way that provides a unique contribution to the field. I thought the authors did a good job of 'considering the options' starting at line 259 ('Shelf sediments as an overlooked sink'). Budget parameters were often determined in 2 ways to support the conclusions. Implications and 'what ifs' were discussed, which shows that the authors have both considered potential issues with higher sinks and, if real, what higher sinks mean for this field and beyond (i.e. paleo-interpretations, etc.). If the larger sink for U is real and the system is not in balance for U, this is very important for the oceanographic community to know.

I have a few broad comments/concerns that would be helpful for the authors to clarify (whether in responses or in the manuscript text itself). In my opinion, addressing these questions/areas would help to strengthen the conclusions and assuage doubts.

Reply 7: We thank the reviewer for the very positive evaluation and constructive comments that help to strengthen the conclusion of our study. We have carefully considered the comments/questions in the revised manuscript, including clarifying the timescale of the sampling, compiling the sedimentary and diagenetic variables to

constrain the representativeness of the fluxes used for the estimation of global shelf sinks, clarifying the uncertainties for a more comprehensive understanding of our estimation. Please find the details below.

1. Potential variability of the datasets used (temporal questions) and how representative the dataset is of global shelf conditions for depths <200 m.

One thing that I would want to see (without looking at references) is what is the timescale of the summer and winter sampling in Figure 1 and Table 1? Could this go in the caption of Fig 1? Essentially, were all the samples from summer taken the same day? Or the same week? The reason this is of interest is to see how variable the Ra/Th could be (averaging over 10 days) from week to week or month to month on the shelf. Would we see much much higher or much much lower a few weeks later? In short, are these representative conditions on this shelf throughout most of the summer and winter? How does the '10 day average' for these radioisotope ratios change at one location over the entire summer? Are there temporal studies in the literature that can be cited here to indicate this range? Two summer dates are shown, although they aren't from the same locations (Fs are down at the bottom of the map and E/Y are in the middle). The results do seem comparable on the scale in Fig. 2. Since there is so much variability, as noted in lines 166-167 (when comparing to other studies) and Fig 1/Fig 5, it would be helpful to know how changeable the Ra/Th ratios are.

Reply 8: We agree that the timescale of Ra/Th is an important issue to consider. We performed sampling campaigns within 5 days (2018 and 2021) or 10 days (2017). Unfortunately, time-series measurement within the timescale of a week/month was not performed in this study. However, no significant inter-annual variation of Ra/Th (Response Figure 1) and ^{224}Ra fluxes (2230 ± 210 vs. 2260 ± 340 dpm $\text{m}^{-2} \text{d}^{-1}$; Cai et al, unpublished data) between the summer of 2019 and 2020 was observed in the study area. In addition, a time-series study of the total ^{224}Ra activities and fluxes in the German Wadden Sea showed good agreement over 12 days, although it is not in the study area of our research (Cai et al., 2020). The results suggest that our estimates based on the $^{224}\text{Ra}/^{228}\text{Th}$ disequilibrium approach are representative of an entire season. We have added the timescale in the figure captions (Fig. 1 in Line 494 and Fig. S2).

Response Figure 2. Comparison of Ra/Th ratios between the sampling campaigns of 2019 and 2020 at St. C12 (122.74°E, 30.94°N) in the East China Sea shelf (Cai et al, unpublished data).

References:

Cai et al. Measurement of ^{224}Ra : ^{228}Th disequilibrium in coastal sediments using a delayed coincidence counter. *Marine Chemistry* 138-139 (2012), 1-6.

Cai et al. Carbon and nutrient export from intertidal sand systems elucidated by $^{224}\text{Ra}/^{228}\text{Th}$ disequilibria. *Geochim. Cosmochim. Acta* 274 (2020), 302-316.

Hong et al. Benthic fluxes of metals into the Pearl River Estuary based on $^{224}\text{Ra}/^{228}\text{Th}$ disequilibrium: From alkaline earth (Ba) to redox sensitive elements (U, Mn, Fe). *Geochimica Cosmochimica Acta* 237 (2018), 223-239.

2. Potential variability of the datasets used (spatial questions) and how representative the dataset is of global shelf conditions for depths <200 m.

Lines 63-68: 'The contrasting conditions...the global oceanic Re and U budgets.' I agree that it's a good region to look at different patterns because of summer/winter changes, my

question would be do most shelves globally look like this one and have similar processes? Do we know or have an idea of how representative they might be? When the estimates are extrapolated to the global model, how does this shelf compare to the typical conditions of the 'average' shelf?

Reply 9: We thank the reviewer for raising this issue. It is worth noting that seasonal variations of Re fluxes and the coupling with organic carbon decomposition were also observed in the North American margins (see Line 289-292): “A similar correlation between sedimentary reductive removal of Re and OC decomposition was observed in the NAMs sediments, overlain by oxygenated bottom waters^{14,35}, suggesting a potentially universal linkage between these two processes in shelf sediments.”

In fact, the sedimentary and diagenetic variables (sedimentation rate, OC supply/decomposition; supplementary Table S3) that dominate the benthic Re and U fluxes in the study fall within the range of the global 'average' shelf. Therefore, we suggest that the fluxes estimated in our study are representative of the average global shelf. We have added the following information (Line 309-314):

"For the evaluation of the sedimentary reductive removal fluxes of Re and U over global shelves, the ECS, the NAMs, and the East Siberian Sea can be adopted as representative settings since they are among the widest shelves worldwide and characterized by diagenetic conditions (sedimentation rate, and OC supply/decomposition) close to the representative conditions of global shelves (Supplementary Table S3)."

If I look at Fig 3 and Fig 2, two of the really high flux points are northern winter points. Do we know (I assume yes) about the shelf dynamics here and can we attribute these patterns to summer vs. winter and not north shelf vs. south shelf spatial trends? Is there more upwelling in one location vs. other? Are the sediment types similar? Are the regressions in Fig 3. still strong (>0.5) if the two northern points are taken away? I realize these measurements are very difficult and occupying the same locations twice might not have been possible. There is some overlap around 31 in summer and winter points. A short comment/discussion from the authors on how the patterns/trends are clearly related

to season and not spatial variability would be welcome (especially for those that don't know the physics and sedimentary systems of this area well).

Reply 10: We concur with the reviewer that shelf dynamics is important for the understanding of the variation. The sampling sites are located in muddy sediments and have relatively homogeneous TOC ($0.33\pm 0.05\%$; Wei et al., 2022). Upwelling in the study area peaks in July and August (summer season) and vanishes in later September (Chen et al., 2021; Hu and Wang, 2016). However, elevated Re and U fluxes are observed in the winter. In addition, there are no spatial patterns related to upwelling (see Response Figure 2). Therefore, we attribute the high fluxes to seasonal rather than spatial variation.

We have added the contrasting dynamics for a better understanding:

Line 92-97 in the Introduction: "The shelf dynamics are distinctly stronger in winter than in summer³⁰: the eddy diffusivities in the bottom water are 5-fold higher; the sediments are more intensively reworked/irrigated as evidenced by deeper penetration of DO and excess ²³⁴Th activity, as well as ²²⁴Ra/²²⁸Th disequilibria. Additionally, upwelling develops in June, reaches its strongest condition in July and August, and eventually vanishes in late September³¹." (Response Figure 3).

Line 229-235 in the Discussion: "Sedimentary OC decomposition dictates the reducing conditions conducive to the removal of Re and U^{13, 14}. Intense winter reworking/irrigation was posited to be responsible for the highly efficient decomposition of sedimentary OC in the study area, resulting in relatively homogeneous TOC contents ($0.33\pm 0.05\%$) and stronger reducing conditions³⁰. Therefore, we argue that the distinctly higher winter removal rates of Re and U are driven by intensive reworking/irrigation."

We have renewed the calculation of the fluxes. Only the stations with smooth removal patterns of porewater Re and U were calculated. Therefore, the northern sites with zig-zag patterns (Re and U at St. B3-W; only U in St. A3-W) have been excluded. For Re, the regressions in Fig. 3 would decrease if St. A3-W was removed. However, the *p* values are generally ≤ 0.05 (see Response Figure 4), and the slope and intercepts generally agree with the original ones within the errors. Therefore, these complications do not challenge our evaluation of the global fluxes and the main conclusions.

Response Figure 3. Sampling locations with the frequently observed upwelling zones (purple polygons) (upwelling zones summarized by Hu & Wang 2016; Chen et al., 2021).

Response Figure 4. Seasonal changes of shelf dynamics at two nearby stations (Y5-S during summer and S1-W during winter): lower $^{224}\text{Ra}/^{228}\text{Th}$ ratios (B) in the sediment column and the stronger eddy diffusivities in the water column (D) indicate stronger shelf dynamics in winter. Figures are taken from Wei et al. (2022)

Response Figure 5. Correlations between sedimentary reductive removal of Re and U versus sedimentary geochemical factors (a,b: amplification factor of sediment area; c,d: the anaerobic decomposition rate of organic carbon determined as the net ammonium production; e,f: sediment oxygen consumption rate) in the East China Sea without the high fluxes from the northern sites.

Reference:

Chen et al. Impact of upwelling on phytoplankton blooms and hypoxia along the Chinese coast in the East China Sea. *Marine Pollution Bulletin* 167 (2021), 112288.

Hu J. & Wang X. H. Progress on upwelling studies in the China seas. *Reviews of Geophysics* 54 (2016), 653-673.

Wei et al. Winter mixing accelerates decomposition of sedimentary organic carbon in seasonally hypoxic coastal seas. *Geochimica Cosmochimica Acta* 317(2022), 457-471.

Line 286 (for U) and early for Re use the median flux extrapolated over the area of global suboxic sediments. How realistic is the resulting estimate? Are all shelves expected to be like the ECS or said in another way, if we know shelves are dynamic/different what are the bounds on these estimates? I agree that the locations in Fig 4 are covering some of the ocean's broadest shelves, however, this doesn't mean that these conditions will be everywhere. A lot of the Hudson Bay numbers look very low compared to the ECS. In short, how homogenous can we expect the shelves to be (i.e. if there is a mix of types, how many are going to have the 'interesting' summer/winter conditions seen in the ECS)? Is there a way to characterize the shelves globally with different types (broad categories) instead of just applying an average or median of the available data to the entire shelf area globally?

Reply 11: Currently, estimation of Re and U fluxes over different seasons have only been conducted in Hingham Bay (Morford et al., 2007), Buzzards Bay (Morford et al., 2009), and the ECS (this study). All the studies show distinct seasonal variations. Indeed, as noted in **Reply 9**, the compiled data are representative of global shelves because their sedimentary and diagenetic parameters that may regulate the Re and U fluxes fall within the representative range of global shelves. We have revised the text to clarify that extrapolation of the average/median would bias the estimate and result in significant uncertainties (with 95% confidence intervals reported). We further stressed that the global shelf fluxes with narrower uncertainties can be estimated based on the correlations

between the reductive removal of Re and U and the sediment oxygen consumption, which is the basis of the later discussion of the implications and 'what ifs':

“It is worth noting that the reductive removal rates of Re and U in suboxic seafloor sediments are expected to drop with water depth due to the less reducing conditions associated with lower OC contents and DO consumption^{13, 14, 50}. Because of the heterogeneity of the fluxes on the global shelves (Supplementary Table S4), simply extrapolating the compiled fluxes to global suboxic sediments may bias the sinks toward the high values of shallow water regions and result in significant uncertainties. We have identified strong correlations between the reductive removal of Re and U and the sediment oxygen consumption due to their synergistic effect during OC decomposition (see last section). The sediment oxygen consumption has been well constrained over the global ocean, with mean areal rates of 25 ± 13 and 9 ± 5 mmol m⁻² d⁻¹ on the global inner (10–50 m) and outer (50–200 m) shelves, respectively⁵⁰. Thus, a more reasonable evaluation of the global shelf sinks of Re and U with narrower uncertainties can be made by taking advantage of the identified correlations between Re and U fluxes and the sediment oxygen consumption rates.” See Line 326-339.

“Based on these updated constraints on the shelf sediment sinks using the correlations between the reductive removal of Re and U and the sediment oxygen consumption, both the Re and U budgets are imbalanced, i.e., the sink is ~80–100% larger than the source (Table 2). Therefore,.....” See Line 353-356.

References:

- Morford J. L., Martin W. R., Kalnejais L. H., François R., Bothner M. and Karle I.-M. Insights on geochemical cycling of U, Re and Mo from seasonal sampling in Boston Harbor, Massachusetts, USA. *Geochim. Cosmochim. Acta* 71, 895-917.
- Morford J. L., Martin W. R., François R. and Carney C. M. (2009) A model for uranium, rhenium, and molybdenum diagenesis in marine sediments based on results from coastal locations. *Geochim. Cosmochim. Acta* 73, 2938-2960.

Along these same lines, how are the 'uncertainties' calculated, or errors extrapolated? I see data from this study (with errors) and data from the literature (no errors) in Supp. Table 4. These were the data used to get the budget estimates starting at Line 259. Medians were reported in this section, but ranges and standard deviations might be helpful too. This could go in Table 2. I'm not sure what the +/- in this table mean (i.e. what do they represent?).

Reply 12: As noted above, we have reported the medians and 95% confidence intervals of the budgets estimated by extrapolating the compiled fluxes in the revised manuscript (Line 315, 3119, 321, 324, 539; and Supplementary Table S4). We emphasize that more reasonable estimations with narrower uncertainties can be made using the correlations between the benthic Re and U fluxes vs. total sediment oxygen consumption rates (L332-339).

The uncertainties of our estimation were propagated from the errors associated with the correlations (this study) and the sediment oxygen consumption rate (Jørgensen et al., 2022). This explanation was added to Table 2 (Line 536-538) and Table S4. Explanations of the uncertainties associated with the other reported data or presented figures were also added (e.g., L523-525, L541-542). For the literature data, most were initially not reported with error.

References:

Jørgensen B. B., Wenzhöfer F., Egger M. and Glud R. N. Sediment oxygen consumption: Role in the global marine carbon cycle. *Earth-Science Review* 228 (2022), 103987.

Additional Line by Line Comments:

From Line 9, it sounds like the authors (from the literature) didn't anticipate shelves being a sink for Re previously, and so that with this study there is now an extra sink for Re (i.e. if the shelf sink is comparable to suboxic/anoxic sinks that were found previously then the total sink has now increased...because otherwise things would still be the same). Just clarifying here that this is what the authors intended: 'Our extrapolation suggests the shelf

sinks are comparable to (for Re) or higher than (~4-fold for U) previous estimations of the suboxic/anoxic sinks. These results suggest that the modern budget of Re and U may be imbalanced and/or their sources are substantially underestimated.'

Reply 13: We have now clarified that the shelf sediment sinks "were previously dismissed or treated as authigenically neutral" (Line 29-30). We reworked the Introduction to clarify that previous estimates were performed in the suboxic and anoxic seafloors beyond the shelf whereas the role of shelf sinks has been dismissed or treated as authigenically neutral (Line 69-75). We further modified the Discussion to ensure that the key finding of the positive correlations between the reductive removal fluxes versus oxygen consumption and the overlooked shelf sediment sinks in the Re and U budgets will be conveyed more clearly to the readers:

Line 332-339: "We have identified strong correlations between the reductive removal of Re and U and the sediment oxygen consumption due to their synergistic effect during OC decomposition (see last section). The sediment oxygen consumption has been well constrained over the global ocean, with mean areal rates of 25 ± 13 and 9 ± 5 mmol m⁻² d⁻¹ on the global inner (10–50 m) and outer (50–200 m) shelves, respectively⁵⁰. Thus, a more reasonable evaluation of the global shelf sinks of Re and U with narrower uncertainties can be made by taking advantage of the identified correlations between Re and U fluxes and the sediment oxygen consumption rates."

Line 353-356: "Based on these updated constraints on the shelf sediment sinks using the correlations between the reductive removal of Re and U and the sediment oxygen consumption, both the Re and U budgets are imbalanced, i.e., the sink is ~80–100% larger than the source (Table 2). Therefore,....."

We have also improved the figures (Fig. 3 and Fig. 6) to improve the readability.

Line 38: 'In fact, the established sinks were based on limited data from continental margins of water depths >300 m, 8.' Based on a quick read of the text, I am assuming that source 8 (Dunk et al, 2022) uses the location in Table 1 to make the shelf sink estimates. I see at least two locations where water depths are <300 m, which would contradict what is said in lines 38-40 of the manuscript. I believe Saanich Inlet is

maximum 230-250m and most depths from the Walvis Bay study (Veeh et al) are 119-225m. It would be helpful if the authors could clarify this point regarding the depth range cutoff used. If Table 1 is the source of the estimates being referred to in source 9, then there seems to be a range in settings and depths (vs. a limited one). I could be mistaken how Table 1 is used, but since this is an important point to the significance of this study (i.e. showing that shallow sediments could be greater than assumed), this part of the text about previous estimates should be very clear.

Reply 14: We concur with the reviewer that a clear description of past estimates is essential. We define the shelf as the setting with water depth ≤ 200 m following the classical definition (e.g., Jørgensen et al. 2022) (Line 72). We have confirmed that the literature data in previous budget evaluations did not include the shelf. Dunk et al. (2002) applied the data in their Table 1 (including the data from the Saanich Inlet and the Walvis Bay) to the anoxic sediments located >200 -m water depth, following Veeh (1967), who stated that "*the effect of Pleistocene sea level fluctuations as a controlling factor in determining the concentration of uranium in sea water is diminished by the occurrence of extensive areas of uranium deposition below 200 m and by the very long residence time of uranium in the ocean*". Indeed, Dunk et al. (2002) also estimated the suboxic flux below 1000-m water depth using the sites from the continental slope, as they stated, "*More recent model results suggest that sediments with Z_{O_2} of 1 cm or less cover only 4–6% of the seafloor, although this is limited to sediments below 1000-m water depth (Morford and Emerson, 1999). Here, we use an area of $6 \pm 2\%$ of the ocean floor ($21.6 \pm 7.2 \times 10^{12} \text{ m}^2$) to obtain an R_s of $13.0 \pm 7.8 \text{ Mmol/year}$ ". We have now clarified that "The established sinks were based on the extrapolation of limited data (collected at sites below ~ 100 to thousands of meters) to the suboxic and anoxic seafloors at water depths $>200 \text{ m}$ ^{3, 19}. " (L69-71).*

References:

Dunk et al. A reevaluation of the oceanic uranium budget for the Holocene. *Chemical Geology* 190 (2002), 45-67.

Morford J. L. and Emerson S. The geochemistry of redox sensitive trace metals in sediments. *Geochimica et Cosmochimica Acta* 63(1999), 1735-1750.

Veeh H. H. (1967) Deposition of uranium from the ocean. *Earth and Planetary Science Letters* 3 (1967), 145-150.

Lines 153, 156 and later uses of 'amplification': Please explain what is meant here by 'extreme amplification of sediment area' and define this term – I had to go to source 18 to get information about the anthropogenic influences. Since amplification is also mentioned in a different context in Line 188, the term should be defined to account for a wider audience reading *Nature Communications*. My question here is also, do we anticipate the amplification factor due to anthropogenic disturbance to be common? If this is not going to happen in most locations on most shelves, then omitting is fine. The authors don't need to add a huge amount of detail, just indicating how anomalous this is expected to be would help rule this out for use in any extrapolation.

Reply 15: Thanks for the helpful suggestions. We have now added the definition and calculation of the amplification factor in the revision:

“The amplification factor of sediment area (see the Method), a dimensionless parameter, is used to characterize the enhancement of solute exchange driven by irrigation relative to that driven by molecular diffusion³⁰.” See Line 202-204.

“Amplification factor (ξ) is then defined as³⁰: $\xi = \frac{F_{Ra}}{\phi \times D_s^{Ra} \times \partial C^{Ra} / \partial z}$. The

calculated amplification factor represents the mean state of irrigation within the upper few cm where ²²⁴Ra/²²⁸Th disequilibria occur.” See Line 461-464.

The St. E1-W, with an extreme amplification factor, is located near a refuge harbor and influenced by accidental anthropogenic disturbance. Other stations show ordinary amplification factors and, thus are not expected to suffer from anthropogenic disturbance.

Fig 1: Overall it's a bit small to read but I like the setup here and it helps the reader quickly understand the Ra/Th and Re, U data. The grey bar is helpful, as are the caption details.

Reply 16: We have now divided each parameter into 5 groups according to the season and the vertical patterns of Re and U (Scenario 1: decreasing downcore; Scenario 2: elevated concentrations in the upper 1 to 2 cm and decreasing downcore; Scenario 3: zig-zag patterns in the upper 6-10 cm). We only present the porewater Re and U concentrations and the Ra/Th ratios of the total sediment in Fig. 1 (Line 494) while moving the other parameters (Fe, Mn, and nutrients) to Supplementary Fig. S2. This revision would make the vertical profiles more easily seen (also a concern of Reviewer #3).

Fig 2: Good visual. Liked it.

Reply 17: Thanks.

Fig 3: Some overlap of text with the points on (e) but overall this was a readable figure. Well structured.

Reply 18: Thanks for pointing it out. We have modified the figures (Line 505).

Fig 4: Good figure. If you needed to cut anything, (a) could be removed. What I was hoping to see is the shelf variability in b/c/d (and that was shown). You can see the shelf depths much more clearly in b/c/d and so those panels would be enough for me with a caption indicating (as is already done) where each is from.

Figure 5: Overall this figure is a good representation of the data and is important. However, the sideways text is a bit confusing/messy and I would recommend using a bracket of some kind at the top or bottom of the plots to indicate data from this study (left, use bracket to cover the red/blue) and the data from the referenced sideways text studies (red/green/purple).

Figure 6: Same comment as fig. 5. The figure will look more professional without the squished/sideways text.

Reply 19: These three figures have been modified as suggested (Line 513, 516 and 521).

Reviewer #3 (Remarks to the Author):

The importance of shallow shelf sediments on the mass balance for U and Re requires more data and closer study, as both are used for reconstructing past anoxic conditions. U has been used more than Re for this purpose, but this is primarily due to the paucity of available Re data. The authors further our quantitative understanding of the importance of irrigation through using $^{224}\text{Ra}/^{228}\text{Th}$ disequilibria, which is particularly important in these shallow shelf sediments. Overall, I would like to see this manuscript published. We use U extensively (both U concentration in sediments and U isotopes in carbonates), which requires understanding sinks and sources, while Re has the potential to be an extremely valuable tracer, but only if more data is obtained from diverse marine locations. The unexpected greater removal in winter compared to summer is also a surprising twist that will encourage the community to consider the seasonal changes in these shallower sediments in different ways. I do have a few minor comments that the authors should consider.

Reply 20: We thank the reviewer's support of our work. We have carefully addressed the comments. Please find our replies below.

- Lines 45-46: The authors suggest that Re and U concentration and isotopic composition are 'emerging proxies', but that statement suggests a recentness that may not be appropriate. U concentrations and isotopic reconstruction of past conditions have been used more extensively than indicated in refs 2-5, and it is for exactly this reason that a better understanding of sinks/sources is required.

Reply 21: Thanks for the constructive comment. We have rephrased the expression: "valuable but still developing proxies". We have also rationalized the references here, although we cannot cite all the work within the reference count (Line 45-47).

- Lines 59-60: the authors cite a lack of Re isotope data, but in 2020 Dickson et al started this process (The rhenium isotope composition of Atlantic Ocean seawater, GCA 287),

along with Dellinger et al. in 2021 (Fractionation of rhenium isotopes in the Mackenzie River basin during oxidative weathering, EPSL, 573).

Reply 22: Thanks for bringing attention to these works. We have changed the statement to "Due to the scarcity of available isotope data^{22, 23, 24}, the Re isotope systematics in the modern ocean has been poorly evaluated." with the references added (Line 65-66).

- Lines 91-93: The authors suggest that authigenic accumulation rates are not used for Re and U due to the high detrital contribution that could hinder the accuracy of the accumulation rate. But is this the case for Re, which has an astonishingly small detrital concentration? More support for this statement would be appropriate.

Reply 23: We agree. During revision, we have removed these sentences as the authigenic accumulation data have also been compiled for comparison. Overall, these data have helped strengthen our conclusion.

- Figure 1: I found it extremely difficult to see the pore water profiles due to the volume of data presented on this figure. I ask whether this might be two figures with some of the profiles separated for each season so that they can be more easily seen, or whether the authors want to consider averaging some of the data and presenting a standard deviation for that profile.

Reply 24: Thanks for raising this issue. We have now divided each parameter into 5 groups according to the season and the vertical patterns of Re and U (Scenario 1: decreasing downcore; Scenario 2: elevated concentrations in the upper 1 to 2 cm and decreasing downcore; Scenario 3: zig-zag patterns in the upper 6-10 cm). We only present the porewater Re/U concentrations and the Ra/Th ratios of the total sediment in Fig. 1 (Line 494) while moving the other parameters (Fe, Mn, and nutrients) to Supplementary Fig. S2. This would make the vertical profiles more easily seen (also a concern of Reviewer #2).

- Line 133: The authors cite the possibility of colloidal phases to explain incomplete

removal, but other ideas instead center around a minimum U concentration required for biological mediation of its removal which should be considered (Lovely et al., 1991, Nature, 350).

Reply 25: Thanks for pointing out this possibility. We have now included it: "Incomplete removal of Re and U from porewater may be associated with a colloidal phase¹² and/or a minimum U concentration required for microbial-mediated removal⁴⁴."(Line 142-144).

Minor comments:

- Line 73: Rather than 'inspires' I would suggest 'causes'.

Reply 26: The wording was corrected as suggested (Line 79).

- Line 166: The decay constant for ²²⁴Ra didn't appear in my version of equation 1, so it was unclear if there was a formatting issue.

Reply 27: The decay constant for ²²⁴Ra is included in the term F_{Ra} . We have now clarified the calculation of F_{Ra} : "where F_{Ra} denotes the flux of total ²²⁴Ra ($\text{dpm m}^{-2} \text{d}^{-1}$) and is calculated as the product of the decay constant of ²²⁴Ra ($\lambda_{Ra} = 0.189 \text{d}^{-1}$) and the integration of the deficit of total ²²⁴Ra (A_{224Ra}) relative to ²²⁸Th (A_{228Th}):

$$F_{Ra} = \lambda_{Ra} \int_0^z (A_{228Th} - A_{224Ra}) dz^{64}.$$

The deficit (²²⁴Ra/²²⁸Th ratio <1) of the solids was confined within the depth of ~10 cm, although they could be scattered due to inhomogeneous sediment particle mixing (Fig. 1c). We here integrate the upper 10 cm to derive the total ²²⁴Ra depletion fluxes. " (Line 456-461).

- Line 342: Should be 'lowering' instead of 'lowing'

Reply 28: Corrected (Line 380).

- Table 2: The percentage symbols can be removed from within the table since they are in the column heading.

Reply 29: Accepted (Line 531).

Reviewer #1 (Remarks to the Author):

Dear editor, dear authors

I would like to thank the authors for their efforts to consider the criticism in the last round of reviews.

However, I am sorry, but I have to criticize again and will therefore unfortunately suggest a further revision or forwarding the manuscript to another journal.

My main criticism relates to what is now the last section of the article (Shelf sediments as an overlooked sink of global oceanic Re and U). While it is debatable whether shelf sediments are an overlooked global sink (see e.g. article: Redox-sensitive trace metals as paleoredox proxies: A review and analysis of data from modern sediments by Bennet and Canfield, which clearly shows that shelf sediments have high Re enrichment, implying that shelf sediments have long been recognized as an important Re sink), my concerns are not related to whether or not shelf sediments are recognized as an important Re sink. Rather, my concerns relate to the lack of recognition of Re flux from the sediments (although the authors have since admitted that it is important and have added a paragraph addressing this issue).

Why did the authors neglect the Re flux from the sediment in their mass balance calculations? Did authors try to take this into account? If so, what is the result of the sum of the reductive Re uptake and the Re flux from the sediment? If the Re flux from the sediment is taken into account, the main conclusion of this article (as far as I understand it) is still the same? That is, modern Re budgets are then still unbalanced and/or their sources are significantly underestimated?

I am sorry but I hope that authors understand that this is an important issue and therefore I have no other choice than suggest another review round or redirection of the manuscript into some other journal.

Reviewer #2 (Remarks to the Author):

I very much appreciated the professional tone of the authors' responses and the detail with which they responded. I think the authors responded well to all the reviewers' comments, not just my own. Figures 5 and 6 now look great and the improvements here really help convey the message better. The additional figures/tables and updates were well chosen to support their argument (and provide direct data responses to the reviewers). I still have a few lingering questions that could only be solved by additional sampling or modeling efforts (e.g. repeated time series work at the same location, as R1 noted higher resolution sampling in the upper 2 cm, and additional modeling work to evaluate the scenarios in S5), but in my opinion, I think the authors have provided a convincing argument with their data as is. Importantly, their presentation in this manuscript leaves room for future refinements and possibilities (e.g. the paper has an open vs. a defensive one) and I will again note that they discussed caveats and 'what ifs' in a way that strengthens their arguments. The authors continue to present the data in a way that advances the field and peaks curiosities, while encouraging others to investigate further.

One minor question/edit:

Supplemental Figure 4 (S4) notes vertical bars for the crustal ratios. I don't see vertical bars in my pdf. I only see points and then a figure label at the bottom of each plot. Perhaps there was a transfer error when the figure was converted to a pdf?

Reviewer #3 (Remarks to the Author):

I am writing in response to the manuscript revisions for "Overlooked shelf sediment reductive sinks of dissolved rhenium and uranium in the modern ocean" by Hong et al. This paper continues to be an important addition to the literature and will spur the community to better evaluate shelf sediments as a sink for Re and U and/or re-evaluate sources to obtain balanced budgets. The authors made compelling and helpful changes in this current revision that have clarified their manuscript and strengthened their argument. I have only a few remaining suggestions:

- In Figure S3, the authors provide the water column Re and U data. Because the one circled data point is not elevated in both Re and U, the sentence should be modified to "The elevated (Re) and lower (U) concentrations (circled) occurred at St. E1-W..."
- Figure S3. Consistent with the units from Fig S7, I am assuming that the units for the U concentration in Fig S3 are nM and not μM .
- The pore water U profiles from Summer 2021 are still confusing to me. I am not sure there is a flux into sediments for F2-S and F4-S the way it is drawn with the red lines on Fig S7. Wouldn't it make more sense to look at the gradient across the sediment-water interface between the bottom water concentration (Fig S3) and the first porewater concentration? If the slope is effectively the same as what you have already determined, then it would be a minor but important footnote.
- I would adjust the caption to Table S3 to be more specific for what you mean by "Globe" by adding "Comparison of key environmental variables between globally representative shelf conditions and the regions chosen for extrapolation."
- In Table S5, the authors include the CRM results. However in Text S1 lines 29-31, the authors also mention measuring MESS-4 as a reference standard. Please include the MESS-4 results, as there is a paucity of Re concentrations published for CRM.
- Figure 3. The authors mention that the red lines in panels e and f denote a regression with zero intercept but I am not seeing any red lines on the plot...?

Response to Reviewers' comments

We thank the editor and reviewers for the constructive comments on our manuscript. We have revised the manuscript to address the issue of the Re and U flux out of the sediments based on the comments. Please see our point-by-point response below.

Reviewer #1 (Remarks to the Author):

Dear editor, dear authors

I would like to thank the authors for their efforts to consider the criticism in the last round of reviews.

However, I am sorry, but I have to criticize again and will therefore unfortunately suggest a further revision or forwarding the manuscript to another journal.

My main criticism relates to what is now the last section of the article (Shelf sediments as an overlooked sink of global oceanic Re and U). While it is debatable whether shelf sediments are an overlooked global sink (see e.g. article: Redox-sensitive trace metals as paleoredox proxies: A review and analysis of data from modern sediments by Bennet and Canfield, which clearly shows that shelf sediments have high Re enrichment, implying that shelf sediments have long been recognized as an important Re sink), my concerns are not related to whether or not shelf sediments are recognized as an important Re sink.

Rather, my concerns relate to the lack of recognition of Re flux from the sediments (although the authors have since admitted that it is important and have added a paragraph addressing this issue).

Reply 1: We thank the reviewer for agreeing with the general conclusion of our study. Indeed, shelf sediments have long been known to be enriched in Re; therefore, it is not surprising that shelf sediments will play an important role in the modern oceanic budget of Re. The novelty of our study is to quantify the contribution of shelf sediment in the oceanic Re and U budgets, as it has been dismissed or treated as authigenically neutral in previous mass balance studies. See Line 66-69.

Why did the authors neglect the Re flux from the sediment in their mass balance calculations? Did authors try to take this into account? If so, what is the result of the sum of the reductive Re uptake and the Re flux from the sediment? If the Re flux from the

sediment is taken into account, the main conclusion of this article (as far as I understand it) is still the same? That is, modern Re budgets are then still unbalanced and/or their sources are significantly underestimated?

I am sorry but I hope that authors understand that this is an important issue and therefore I have no other choice than suggest another review round or redirection of the manuscript into some other journal.

Reply 2: As stated in the first section of the Discussion, the enrichment of core-top $[\text{Re}]_{\text{diss}}$ is most likely regenerated from the decomposition of freshly deposited organic matter that scavenged Re from seawater rather than from the re-oxidation of the previously-reduced Re and/or riverine particle-bound phases. We have now added a section (The regenerated flux of Re and U out of the sediment–water interface) to discuss the fluxes out of the sediments on the Re and U mass balance. We clarified that these fluxes are small relative to the reductive removal fluxes and will not change our conclusion on the quantitative importance of shelf sediment sinks to the modern oceanic budgets of Re and U. We have added the following text to the Discussion. See Line 239-261:

“The regenerated upward fluxes of Re and U estimated with our porewater data range from -860 to 0 $\text{pmol m}^{-2} \text{d}^{-1}$ (median: -49 $\text{pmol m}^{-2} \text{d}^{-1}$; average: -223 $\text{pmol m}^{-2} \text{d}^{-1}$) and -40 to 0 $\text{nmol m}^{-2} \text{d}^{-1}$ (median: 0 $\text{pmol m}^{-2} \text{d}^{-1}$; average: -6 $\text{pmol m}^{-2} \text{d}^{-1}$), respectively (Table 1). For some sites, the regenerated Re and U fluxes are comparable to or even larger than the sedimentary reductive removal fluxes. Regarding the median values, they are ~15% and 0% of the sedimentary reductive removal fluxes of Re and U, respectively, since many sites have zero upward fluxes. Regarding the average values, the regenerated upward fluxes could be ~36% and ~6% of the reductive removal fluxes of Re and U at our studied sites, respectively. It suggests that the regenerated flux could be an important yet not well understood source for the reductive removal of Re and U within the sediments. However, unlike rare earth elements whose riverine particle-bound phases might be actively involved in the shelf cycling (e.g., refs. ^{26, 48, 49}), the regenerated upward flux of Re and U are most likely scavenged from seawater, as discussed above. Therefore, they are considered part of oceanic internal recycling. In this regard, we have not included these fluxes as external sources when discussing the oceanic Re and U mass

balance. Even if some of the upward fluxes are from riverine particle-bound phases and/or re-oxidation of previously reduced species, our conclusion on the quantitative importance of shelf sediment sinks to the modern oceanic budgets of Re and U remains unchanged. Future studies should aim to better characterize the nature of Re and U released at the sediment–water interface, for example, by combined analyses of their isotope signatures and exchange flux. ”

Reviewer #2 (Remarks to the Author):

I very much appreciated the professional tone of the authors’ responses and the detail with which they responded. I think the authors responded well to all the reviewers’ comments, not just my own. Figures 5 and 6 now look great and I the improvements here really help convey the message better. The additional figures/tables and updates were well chosen to support their argument (and provide direct data responses to the reviewers). I still have a few lingering questions that could only be solved by additional sampling or modeling efforts (e.g. repeated time series work at the same location, as R1 noted higher resolution sampling in the upper 2 cm, and additional modeling work to evaluate the scenarios in S5), but in my opinion, I think the authors have provided a convincing argument with their data as is. Importantly, their presentation in this manuscript leaves room for future refinements and possibilities (e.g. the paper has an open vs. a defensive one) and I will again note that they discussed caveats and ‘what ifs’ in a way that strengthens their arguments. The authors continue to present the data in a way that advances the field and peaks curiosities, while encouraging others to investigate further.

Reply 3: We appreciate the reviewer’s very positive evaluation of our manuscript. We agree that more future work, including field sampling (high-resolution sampling of dissolved Re and U across the sediment–water interface) and modeling work to discern controls on the cycling and fluxes in marine sediments, are essential to solving the open questions regarding the processes and associated fluxes contributing to the Re mass balance.

One minor question/edit:

Supplemental Figure 4 (S4) notes vertical bars for the crustal ratios. I don't see vertical bars in my pdf. I only see points and then a figure label at the bottom of each plot.

Perhaps there was a transfer error when the figure was converted to a pdf?

Reply 4: We have removed the vertical bars and the caption in this figure in the revision.

Reviewer #3 (Remarks to the Author):

I am writing in response to the manuscript revisions for “Overlooked shelf sediment reductive sinks of dissolved rhenium and uranium in the modern ocean” by Hong et al. This paper continues to be an important addition to the literature and will spur the community to better evaluate shelf sediments as a sink for Re and U and/or re-evaluate sources to obtain balanced budgets. The authors made compelling and helpful changes in this current revision that have clarified their manuscript and strengthened their argument. I have only a few remaining suggestions:

- In Figure S3, the authors provide the water column Re and U data. Because the one circled data point is not elevated in both Re and U, the sentence should be modified to “The elevated (Re) and lower (U) concentrations (circled) occurred at St. E1-W...”
- Figure S3. Consistent with the units from Fig S7, I am assuming that the units for the U concentration in Fig S3 are nM and not uM.

Reply 5: Thanks for pointing out. We have changed the units and the figure caption.

- The pore water U profiles from Summer 2021 are still confusing to me. I am not sure there is a flux into sediments for F2-S and F4-S the way it is drawn with the red lines on Fig S7. Wouldn't it make more sense to look at the gradient across the sediment-water interface between the bottom water concentration (Fig S3) and the first porewater concentration? If the slope is effectively the same as what you have already determined, then it would be a minor but important footnote.

Reply 6: The concentration gradients are -1.17 for F2-S and -0.95 for F4-S using the bottom water and the first porewater (at 1 cm), and they are broadly consistent with the

estimation in Fig S7 (-1.24 ± 0.14 and -1.73 ± 0.30 ; Table S6) within a factor of ~ 2 . Thus, there should be a flux into sediments for the two sites.

- I would adjust the caption to Table S3 to be more specific for what you mean by “Globe” by adding “Comparison of key environmental variables between globally representative shelf conditions and the regions chosen for extrapolation.”

Reply 7: We have corrected the caption of Table S3.

- In Table S5, the authors include the CRM results. However in Text S1 lines 29-31, the authors also mention measuring MESS-4 as a reference standard. Please include the MESS-4 results, as there is a paucity of Re concentrations published for CRM.

Reply 8: The results of MESS-4 have been added.

- Figure 3. The authors mention that the red lines in panels e and f denote a regression with zero intercept but I am not seeing any red lines on the plot...?

Reply 9: We have removed the sentence because only the regressions with zero intercept are shown (black lines) in panels e and f.

Reviewer #3 (Remarks to the Author):

I appreciate the work that the authors have done to determine the relative importance of the Re and U fluxes into sediments for shelf sediments. The additional calculations to put the flux from sediments into context were important additions. The relative magnitudes of these fluxes compared with riverine sources and anoxic sinks provide evidence for the key issue at hand - whether the sinks and sources are balanced and the subsequent impact on residence times. I recommend publication.

If there remains an additional concern regarding the title of the section that begins on line 325 "Shelf sediments as an overlooked sink of global oceanic Re and U" due to the fact that there has been previous work that highlighted shelf sediments as a relevant sink, the authors could consider rewording this section as "Shelf sediments as an important (or critical or dominant) sink of global oceanic Re and U"

Response to Reviewers' comments

REVIEWERS' COMMENTS

Reviewer #3 (Remarks to the Author):

I appreciate the work that the authors have done to determine the relative importance of the Re and U fluxes into sediments for shelf sediments. The additional calculations to put the flux from sediments into context were important additions. The relative magnitudes of these fluxes compared with riverine sources and anoxic sinks provide evidence for the key issue at hand - whether the sinks and sources are balanced and the subsequent impact on residence times. I recommend publication.

If there remains an additional concern regarding the title of the section that begins on line 325 "Shelf sediments as an overlooked sink of global oceanic Re and U" due to the fact that there has been previous work that highlighted shelf sediments as a relevant sink, the authors could consider rewording this section as "Shelf sediments as an important (or critical or dominant) sink of global oceanic Re and U"

Reply: We have rephrased this section as "Shelf sediments as a critical sink of global oceanic Re and U". See Line 331.